# Machine learning aided construction of the quorum sensing communication network for human gut microbiota

Shengbo Wu [1,2], Jie Feng [3], Chunjiang Liu [1,2], Hao Wu [4], Zekai Qiu [1], Jianjun Ge [1], Shuyang Sun [1], Xia Hong [1], Yukun Li [1], Xiaona Wang [1], Aidong Yang [5✉], Fei Guo [6✉] & Jianjun Qiao [1,4,7✉]

Quorum sensing (QS) is a cell-cell communication mechanism that connects members in various microbial systems. Conventionally, a small number of QS entries are collected for specific microbes, which is far from being able to fully depict communication-based complex microbial interactions in human gut microbiota. In this study, we propose a systematic workflow including three modules and the use of machine learning-based classifiers to collect, expand, and mine the QS-related entries. Furthermore, we develop the Quorum Sensing of Human Gut Microbes (QSHGM) database (http://www.qshgm.lbci.net/) including 28,567 redundancy removal entries, to bridge the gap between QS repositories and human gut microbiota. With the help of QSHGM, various communication-based microbial interactions can be searched and a QS communication network (QSCN) is further constructed and analysed for 818 human gut microbes. This work contributes to the establishment of the QSCN which may form one of the key knowledge maps of the human gut microbiota, supporting future applications such as new manipulations to synthetic microbiota and potential therapies to gut diseases.

---

[1] School of Chemical Engineering and Technology, Tianjin University, Tianjin 300072, China. [2] State Key Laboratory of Chemical Engineering, Tianjin University, Tianjin 300072, China. [3] School of Computer Science and Technology, College of Intelligence and Computing, Tianjin University, Tianjin 300350, China. [4] Zhejiang Shaoxing Research Institute of Tianjin University, Shaoxing 312300, China. [5] Department of Engineering Science, University of Oxford, Oxford OX1 3PJ, UK. [6] School of Computer Science and Engineering, Central South University, Changsha 410083, China. [7] Key Laboratory of Systems Bioengineering, Ministry of Education (Tianjin University), Tianjin 300072, China. ✉email: aidong.yang@eng.ox.ac.uk; guofei@csu.edu.cn; jianjunq@tju.edu.cn

Human gut microbiota is a dynamic and complex microbial system[1] that links to the pathogen colonization resistance[2], immune system regulation[3], and human health maintenance[4]. Recent breakthroughs in high-throughput screening and multi-omics technologies have enabled the detection and quantification of the microbiota composition[5] in the human gut system. More and more research suggests that engineering the gut microbiota and regulating the microbial interactions[6,7] can be viewed as potential novel therapeutics for treating diverse gut diseases[8].

Quorum sensing (QS), a population-level communication mechanism, has huge potential to be engineered for regulating microbial interactions and developing future therapies[9,10]. Generally, there are diverse QS signals termed as microbial languages for intraspecies (N-Acyl-homoserine lactones, AHLs; diffusible signal factors, DSFs; 4-hydroxy-2-alkylquinolines, HAQs; cholera autoinducer 1, CAI-1; auto-inducing peptides, AIPs; dialkylresorcinols; photopyrones)[11,12] and interspecies (autoinducer 2, AI-2; indole) communications[13,14]. The above QS languages in natural microbial systems such as gut microbiota play a significant role in the QS-based interactions, which are closely relevant to various diseases[15]. For example, N-(3-oxodecanoyl)-L-homoserine lactone, a common AHL-type signal, plays an important role in the modulation of the gut immune system by inducing neutrophils apoptosis[16] and attenuating innate immune responses via disruption of NF-kB signaling[17], thus providing better colonization for Pseudomonas aeruginosa in the host. DSF analogs were verified to strengthen the mucosal barrier and reduce antibiotic tolerance of P. aeruginosa[18]. Different hosts can utilize the aryl hydrocarbon receptor (AhR) to "listen in" the concentration of the HAQs from P. aeruginosa to regulate immune responses dynamically[19]. CAI-1 from V. cholerae can be designed to be recognized by an engineered L. lactis specifically in the gut, and the lactic acid from the engineered strain can repress the infection of V. cholerae in turn[20]. AI-2 produced by Ruminococcus obeum could repress several colonization factors of Vibrio cholerae, thus restricting the colonization of V. cholerae, which leads to diarrheal diseases[21]. Furthermore, indole has been confirmed to increase the expression of anti-inflammatory genes, elicit proinflammatory effects, affect the immune system of hosts, and decrease pathogen colonization[14,22]. The evidence stated above suggests that manipulations of the level of diverse QS languages such as AI-2[23] in microbial communication play an important role in diverse host-centric applications for gut microbial systems. Therefore, in our previous study[24], we have proposed the "QS communication network" (QSCN), a unifying concept for vertical and horizontal QS-based interactions implemented through producing, transducing, and responding to QS signaling molecules, to indicate its important role in host-centric probiotic manipulations and various practical applications of synthetic microbial consortia. QSCN calls for a comprehensive QS database, which includes the collections of human gut microbes and QS repository, to bridge the gap between existing QS-related repositories and human gut microbiota.

Some existing databases relevant to gut microbiota or diverse QS systems have been constructed separately to provide data integration and interpretation for relevant research. With respect to the gut microbiota, the gutMEGA database[25] contains thousands of gut microbiota compositions (metagenomic sequences), phenotypes, and experimental information. GMrepo[26] focuses on the annotated human gut metagenomes to facilitate the development of human metagenomic data. BIO-ML[27] includes 7,758 gut bacterial isolates, 3632 genome sequences, and diverse longitudinal multi-omics data. Particularly, VMH[28] is a database that has integrated thousands of metabolites, reactions, human genes, microbes (818 strains), microbial genes, and food items that link

to hundreds of gut diseases and nutritional data. With regard to QS, repositories of limited QS systems in Gram-negative and Gram-positive bacteria have previously been curated to form SigMol[29] and Quorumpeps[30], respectively. P2CS[31,32] was constructed and updated for a two-component system (TCS), which is a typical communicating system that is composed of a histidine kinase receptor and a response regulator partner. Furthermore, we have previously developed the QSIdb database[33] to expand the potential QS interference molecules for different QS systems. We applied a pipeline including SMILES-based algorithms and docking-based validations to obtain a potential QS interference molecules dataset (73,073 compounds) from the existing compounds in the PubChem database. Note that some recent databases such as gutMDisorder[34] have linked the human microbiota and many macro-environmental factors together to describe the intervention and regulation of various diseases. In addition, exogenous active substances and endogenous host factors were also collected for human microbiota into MASI[35] and GIMICA[36], respectively, to provide information on the interactions of various substances and gut microbiota.

While gut microbiota and QS systems have been curated in various databases, they have largely been collected separately so far, which may limit the understanding of communication-based complex microbial interactions in human gut microbiota. Furthermore, existing studies have often focused on using limited reported QS entries; novel QS entries mining and integration to form a relatively complete network is yet to be further explored. Although some biological networks such as metabolite-based interaction networks have been relatively mature, they cannot decipher complex regulatory relationships among microorganisms, thus leading to the incompleteness of microbial interaction networks.

In this study, we aim to address the above deficiencies through a combination of various methods (a framework diagram can be found in Supplementary Fig. 1). We firstly developed a systematic workflow including entry collecting, expanding, and mining modules to construct a QS repository for human gut microbiota. In the collecting module, due to the intricate overlaps on QS and two-component system (TCS) entries, we curated the annotated QS and TCS (QS&TCS) entries carefully for each component in the human gut microbiota to form a repository of reported QS entries as inclusively as possible. Information gathering in this module was also combined with machine learning (ML) algorithms including random forest (RF), k-nearest neighbor (KNN), support vector machine (SVM), and deep neural network (DNN) to develop four classifiers, which were then used in the expanding module to nominate further candidates of human gut QS entries from existing (general-purpose) QS databases. These candidates were finally analysed in the mining module, where protein annotation, functional analysis, and homologous modeling were combined to re-annotate and mine QS entries. These have led to a QS database of human gut microbiota (QSHGM, http://www.qshgm.lbci.net/) including the reported (21,383) and extended (7184) QS entries, which offers user-friendly browsing and searching functions to support various applications. With the help of QSHGM, we can search complex regulation-based interactions for different microbial consortia and further constructed a QSCN to visualize and decipher intricate QS-based interactions for human gut microbiota. Finally, we identified key challenges and suggest directions for the QSCN and how we can engineer them to provide more future applications.

## Results

**The systematic workflow for QSHGM.** We developed a systematic workflow which includes three modules (collecting,

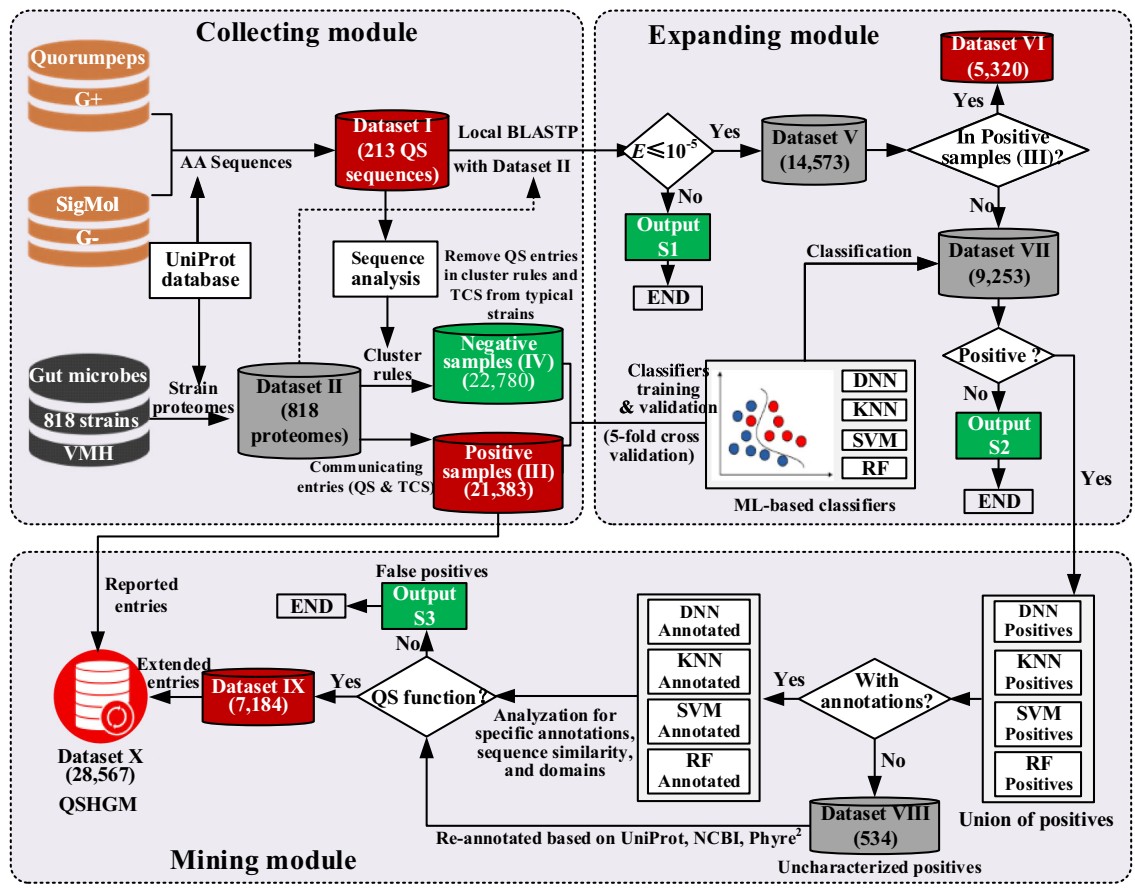

**Fig. 1 Schematic diagram of the systematic workflow including three modules.** There are ten engaged datasets in our systematic workflow, i.e., 213 validated QS entries from Gram-positive (G+) and Gram-negative (G−) microbes (Dataset I) (Supplementary Data 1), 818 proteomes for the gut microbiota from VMH and UniProt (Dataset II, https://pan.baidu.com/s/1o46nn1b7L5nvCqgpwW7Zlw. Password: tfnx), positive samples collected manually from dataset I (Dataset III) (Supplementary Data 2), negative samples obtained from dataset I (Dataset IV) (Supplementary Data 3), results of local BLASTP with E ≤ 10-5 (Dataset V) (Supplementary Data 4), overlaps of the reported QS entries in dataset III and V (Dataset VI) (Supplementary Data 5), proteins dataset excluded dataset VI for dataset V (Dataset VII) (Supplementary Data 6), uncharacterized positives classified by different ML-based classifiers (Dataset VIII) (Supplementary Data 7), extended QS entries (Dataset IX) (Supplementary Data 8), and total data for QSHGM (Dataset X) (Supplementary Data 10). There are another three abandoned datasets in the workflow of the systematic workflow, i.e., protein datasets with E > $10^{-5}$ (Output S1), negative ones classified by ML-based classifiers (Output S2), and proteins without QS functions (false positives) (Output S3, Supplementary Data 9). Details of the above datasets are provided in Supplementary Table 1. Note that positive/negative/mixed datasets are colored in red/green/gray, respectively.

expanding, and mining modules) and four classifiers based on ML algorithms to construct a QS repository for human gut microbiota (Fig. 1). In the collecting module, we firstly obtained 213 recognized QS entries (Dataset I) from SigMol and Quorumpeps databases and curated their corresponding amino acid sequences from the UniProt database. TCS entries play an important role in microbial communications, which overlap with QS, but it is difficult to separate them clearly. In this work, we started by manually searching the 818 gut microbes from the VMH database[28] (Dataset II) to collect reported both QS and TCS (QS&TCS) entries which are termed "positive samples" (Dataset III, 21,383 entries) to cover the reported QS entries as inclusively as possible for constructing a comprehensive microbial communication database. The manual search was based on commonly used QS ("quorum sensing", "LuxR", "tryptophanase") or TCS ("two-component") annotations. The negative samples (Dataset IV, 22,780 entries) were then obtained by removing QS&TCS entries from typical proteomes in Dataset II, such as *Escherichia coli* and *Pseudomonas aeruginosa* (more details in Method section) that conform to QS cluster rules. These rules were developed based on Dataset I through sequence

analysis, including evolution analysis, QS-relevant protein annotations, and amino acid sequence descriptors comparison (more details in Method section). In the expanding module, we obtained an extended dataset (Dataset V, 14,573 entries) from the results of the local BLASTP[37] on Dataset I and II with the criteria of the E value[38] being smaller than 10-5, which is commonly used in the sequence alignment to obtain homologs. Four different ML algorithms (DNN, SVM, RF, and KNN) were used to construct classifiers, which were trained and validated based on the above positive (III) and negative samples (IV) to obtain more potential QS entries. After excluding from Dataset V those which were already collected as the reported QS&TCS entries in dataset III (Dataset VI, 5320 entries), the remaining entries (Dataset VII, 9,253 entries) were then classified by the four ML-based classifiers stated above. The output of these classifiers was further processed in the mining module, where the union of the four positives predicted by the four classifiers were divided into uncharacterized positives (Dataset VIII, 534 entries) and annotated positives. The uncharacterized positives were re-annotated, mined, and sorted out manually with the help of UniProt[39], NCBI (https://www.ncbi.nlm.nih.gov/) and Phyre[2] databases[40]. Furthermore, we

conducted the function analysis by checking their specific annotations, sequence similarity, and domains (see more details in Supplementary Data 11) for the annotated/re-annotated union of positives to decide whether the entry has a QS function (true positives, Dataset IX, 7,184 entries) or not (false positives, Output S3, 438 entries), if so, whether it is a QS synthase or a QS receptor. A combination of manual curation, BLASTP-based expanding, and multiple ML-based classifications helped us obtain as many potential QS entries as possible. Finally, the extended QS entries and the reported QS&TCS entries were combined together to form the QSHGM (Dataset X, 28,567 entries) database (http://www.qshgm.lbci.net/).

**Reported and annotated QS entries**. There are 84 autoinducer synthases and 129 QS receptors in dataset I. With respect to autoinducer synthases, we divided them into seven types, i.e., AHLs, DSFs, AI-2, indole, HAQs, CAI-1, and others. As a result, AHLs synthases account for the vast majority, which among other possibilities can be divided into two protein families, LuxI (from *Vibrio fischeri*) and YenI (from *Yersinia enterocolitica*) (Fig. 2a). With regard to QS receptors, we also divided them into seven types, i.e., LuxR-type, TCS type, CAI-1 receptor, AI-2 receptor, DSFs receptor, HAQs receptor, and other receptors (Fig. 2b). LuxR and TCS type receptors account for the vast majority of QS receptors. Similarly, LuxR-type receptors can be roughly divided into two protein families, LuxR (from *V. fischeri*) and YenR (from *Y. enterocolitica*). Note that the evolutionary trees of AHLs synthases and their receptors counterpart are in a high similarity (Fig. 2a, b), part of which was also identified by Gray et al.[41]. This indicates that there is coevolution for AHLs synthases and their corresponding receptors.

There are 1640, 5921, 66, and 15,703 entries for "quorum sensing", "LuxR", "tryptophanase", and "two-component", respectively (Fig. 2c). LuxR-type and TCS entries account for the vast majority, which are 25.38 and 67.31%, respectively. We have also shown the distribution of QS&TCS entries for each strain based on the seven-strain simplified human microbiomes (SIHUMIs) used by Colosimo et al.[42] (Fig. 2d). This verified that LuxR-type QS and TCS entries account for the vast majority of QS&TCS entries in these strains. Furthermore, we noted that there are certain overlaps in the distribution of the four entries. For example, there are seven entries (P69409, P0ACZ6, P0AGA8, P66798, P0AF30, P0AEL9, and Q8XE66) in the *E. coli* O157:H7 strain (Fig. 2e), which are both LuxR-type and TCS receptors. In addition, we have counted and distributed the total QS&TCS entries of the 818 gut microbes from the VMH database[28] to form a better picture of the QS repository in human gut microbiota (Fig. 2f). According to the cumulative distribution curve for the statistics (Fig. 2f subgraph), we found that about 90% of strains contain less than 60 QS&TCS entries, and only seven strains have more than 150 entries. This distribution will be revisited after extended QS entries are included (see below).

**Expanded QS entries**. The amino acid composition (AAC) calculates the frequency of each amino acid type in a protein sequence. The frequencies of all 20 natural amino acids are the percent of the number of amino acid types divided by the length of a protein sequence[43] (more details in the Method section). We calculated the frequency of each amino acid type in each entry sequence as the protein features, and we conducted a fivefold cross-validation to train classifiers using the positive (Dataset III) and negative samples (Dataset IV), where the average accuracy, prediction, recall, and *F*1 score (more details were listed in method section) were applied to evaluate their performances. The results show that the performances of the DNN, SVM, KNN, and

RF classifiers were not very different, with the RF-based classifier being slightly more prominent (Fig. 3a). We then manually checked the annotations of the predicted results from the classification of the four ML-based classifiers on Dataset VII and divided the four positives into annotated positives and uncharacterized positives (Dataset VIII, 534 entries) (Fig. 3b), which were analysed further for their specific overlaps (Supplementary Fig 2) (see more details in Supplementary Data 12). In order to obtain as many potential QS entries as possible, it was helpful to combine the four positives from the four classifiers together to form a union.

With the help of the functional analysis (Supplementary Data 11), we then re-annotated the 534 uncharacterized entries and grouped them into nine protein clusters manually (Fig. 3c), in which the histidine kinase (a major component in a TCS) occupied the majority. Note that there were another 28 entries that were vaguely described without specific protein annotations (Fig. 3c). As listed in Table 1 and Table 2, these entries were further explored and re-annotated based on the web BLASTP of the NCBI database and Phyre[2], respectively. There were 20 proteins (Table 1) that can be re-annotated based on the BLASTP results from NCBI. Except for U2J6M1 and C0C5Y6, there is much potential for the other 18 proteins to be QS proteins. ArsR, a component of ArsRS TCS, regulates the acid adaptation and biofilm formation of the pathogen *Helicobacter pylori* in the human gut[44]. Beta-ketoacyl-ACP synthase III catalyzes the condensation reaction of fatty acid synthesis, which indicates that there is potential for *Prevotella bivia* to produce Dialkylresorcinols just like the function of DarB from *Photorhabdus asymbiotica*[45]. The histidine kinase, LuxR family regulator, and Rgg/GadR/MutR family regulator are important parts of TCS, LuxR-type, and Rgg-based QS systems[46], respectively.

There were eight entries (Table 2) that have no specific annotations or classifications in NCBI or UniProt database. We submitted these protein sequences to Phyre[2] to investigate the 2D and 3D structures of their models, their domain compositions, and model quality. A0A4Y4IIW5 and A0A5C4P2T9 are signaling proteins and AgrC (belonging to Agr QS system[47]) family proteins, respectively. This indicates that *Lysinibacillus fusiformis* and *Streptococcus salivarius* may have some protein components of the *agr* QS system, thus producing and/or responding to the same QS signaling peptide as common pathogen *Staphylococcus aureus*. The other six of them are templated on the AimR transcriptional regulator, which is the intracellular signal peptide receptor for the QS-based communication between viruses that guides lysis–lysogeny decisions[48]. This suggests that different Bacillus phages may "listen in" diverse bacterial hosts, such as *Bacillus amyloliquefaciens*, *Bacillus mycoides*, *Bacillus thuringiensis*, and *Bacillus atrophaeus*, to coordinate lysis–lysogeny decisions.

Furthermore, we have further conducted the function analysis and checked their specific annotations, sequence similarity, and domains for the annotated/re-annotated union of positives (Supplementary Data 11) to decide whether the entry has a QS function (true positives, Dataset IX, 7184 entries, more details in Supplementary Data 8) or not (false positives, Output S3, 438 entries, more details in Supplementary Data 9). Finally, the reported QS&TCS and extended QS entries were combined together to be Dataset X (28,567 entries). To sum up, with the help of the proposed systematic workflow (Fig. 1), we obtained a comprehensive QS repository including the manually collected 21,383 positive samples (Database III) and the extended 7184 ones (Database IX) for 818 gut microbes, and the total 28,567 entries (Database X) are composed of 1882 QS synthases and 26,685 receptors. There was a 33.60% increase in extended entries (Database IX) for Dataset X (Fig. 3d) from the previous

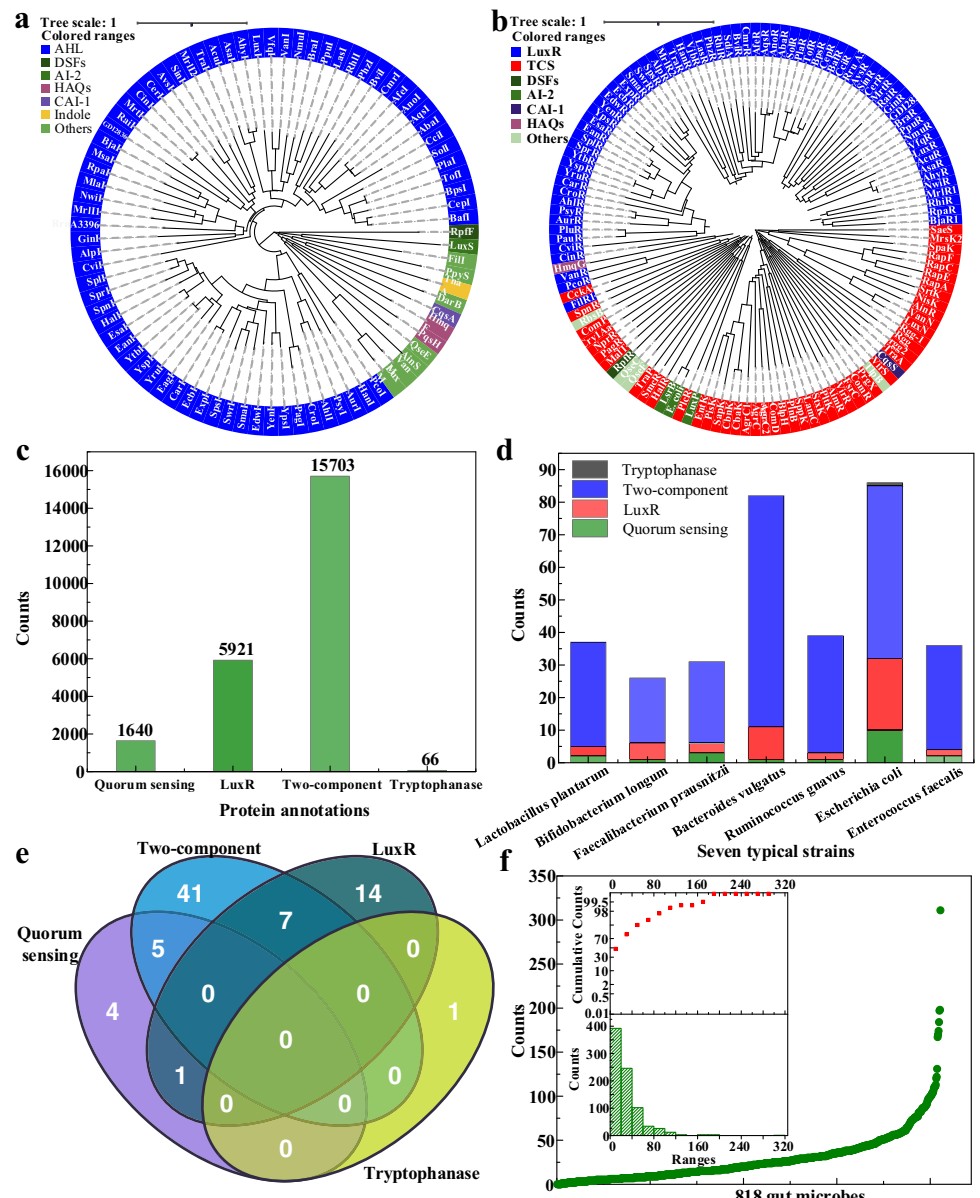

**Fig. 2 Results of collections of the reported and annotated QS entries.** Evolutionary trees of QS synthases (**a**) and receptors (**b**). **a** The optimal tree with the sum of branch length = 40.33 is shown. This analysis involves 84 amino acid sequences, and there are a total of 1374 positions in the final dataset. **b** The optimal tree with the sum of branch length = 91.14 is shown. This analysis involves 129 amino acid sequences, and there are a total of 1010 positions in the final dataset. **c** Total QS&TCS entries with four protein annotations, i.e., "quorum sensing", "LuxR", "two-component", and "tryptophanase". **d** QS&TCS entries distribution of the seven-strain simplified human gut microbes used by Colosimo et al.[42]. **e** The overlap of the four types of QS&TCS entries in *Escherichia coli* O157:H7 strain. **f** QS&TCS entries count distribution of 818 human gut microbes from the VMH database[28]. Note that the subgraph indicates the cumulative distribution curve for the statistics of the collected QS&TCS entries. The upper and lower insets show the probabilities and histogram, respectively. "Ranges" in the subgraph shows the range of the number of QS&TCS entries contained in each strain. The "Counts" and "Cumulative counts" in the subgraph represents the specific number of strains and proportion, respectively. Source data are provided as a Source Data file.

annotation-based collections (Database III) (Fig. 2f). Note that we have mined eight potential QS proteins (Table 2) with the help of functional analysis and homologous modeling, which is of great significance for the further exploration of the related QS mechanism and their applications. To enable user-friendly browsing and searching for entries identified in this work, we constructed a comprehensive QS-related database of human gut microbiota (QSHGM), which is freely available at: http://www.qshgm.lbci.net/. There is a simple user guide for QSHGM browsing and searching (Supplementary Fig 3) in the supplementary information.

**QS-based interactions prediction.** QS-based interactions play an essential role in deciphering complex interactions of natural microbial systems and dynamically manipulating diverse synthetic microbial consortia. The collected data in the QSHGM can enable the prediction of the existence of QS-based microbial interaction by querying whether any pairwise microbes can speak the same QS language. For example, due to speaking the AI-2 language, we predicted AI-2-based communication between *E. coli* O157:H7 and *Bacteroides pectinophilus* ATCC 43243 (Fig. 4a), which is in line with the previously reported observation that AI-2 produced by *E. coli* can influence the Bacteriodetes[49].

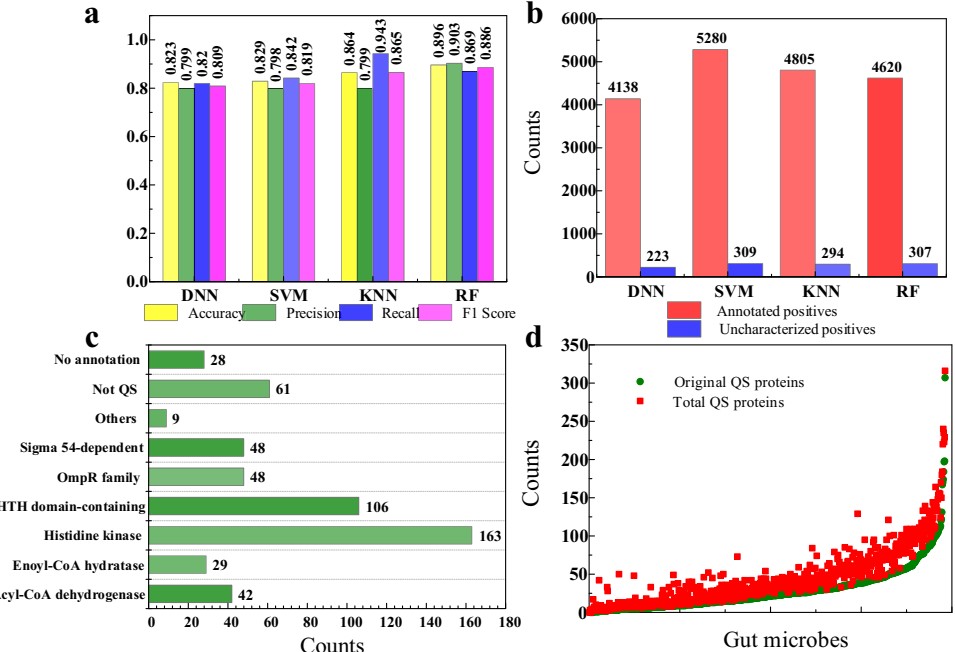

**Fig. 3 Results of the expansion and mining for QS entries based on the proposed systematic workflow. a** Accuracy, precision, recall, and *F*1 score of the four classifiers based on DNN, SVM, KNN, and RF algorithms. **b** Counts of the annotated and uncharacterized positives for the union set of positives from DNN, SVM, KNN, and RF classifiers; **c** Results of the protein clusters of 534 re-annotated protein entries. **d** Distribution total of 28,567 redundancy removal entries in 818 gut microbes. Source data are provided as a Source Data file.

**Table 1 Results of 20 expanded entries without existing annotations based on the web BLASTP.**

| Strains | TaxID | Entry | Template | Query cover | Percent identity | New annotations |
|---|---|---|---|---|---|---|
| *Halococcus morrhuae* | 931277 | M0MA34 | WP_004054989.1 | 100% | 100% | ArsR subfamily of regulator |
| *Clostridium hylemonae* | 553973 | C0C300 | WP_006443816.1 | 100% | 100% | Autoinducer 2 ABC transporter |
| *Prevotella bivia* | 868129 | I4Z9V6 | WP_036847997.1 | 80% | 80.39% | Beta-ketoacyl-ACP synthase III |
| *Enterococcus caccae* | 1158612 | R3TYZ5 | WP_069646785.1 | 100% | 80.80% | Histidine kinase |
| *Lactobacillus ruminis* | 525362 | E7FSN7 | WP_003695050.1 | 98% | 98.96% | Histidine kinase |
| *Streptococcus peroris* | 888746 | E8KCS5 | WP_070888551.1 | 100% | 99.58% | Histidine kinase |
| *Streptococcus parauberis* | 1348 | A0A3E1JFV3 | WP_116486843.1 | 100% | 100% | Histidine kinase |
| *Hungatella hathewayi* | 566550 | D3ADP6 | PXX46370.1 | 98% | 92.45% | LuxR family regulator |
| *Enterococcus cecorum* | 1121864 | S1R0J3 | WP_047242627.1 | 100% | 97.31% | Rgg/GadR/MutR family regulator |
| *Enterococcus cecorum* | 1121864 | S1R7E8 | WP_171336239.1 | 98% | 93.70% | Rgg/GadR/MutR family regulator |
| *Streptococcus constellatus* | 1035184 | U2ZME3 | WP_022525523.1 | 100% | 100% | Rgg/GadR/MutR family regulator |
| *Streptococcus equinus* | 525379 | E8JR85 | WP_029875994.1 | 97% | 97.20% | Rgg/GadR/MutR family regulator |
| *Streptococcus intermedius* | 1095731 | U2XPZ3 | WP_003032153.1 | 100% | 100% | Rgg/GadR/MutR family regulator |
| *Candidatus Melainabacteria* | 2052166 | A0A3S0FWU1 | MBI4533416.1 | 80% | 47.68% | Sensor histidine kinase |
| *Candidatus Melainabacteria* | 2052166 | A0A431KQ57 | MBI5174129.1 | 79% | 47.28% | Sensor histidine kinase |
| *Coriobacteriales bacterium* | 2491116 | A0A437UTJ5 | WP_130811315.1 | 99% | 43.81% | Sensor histidine kinase |
| *Lactobacillus amylolyticus* | 585524 | D4YTV9 | EST03116.1 | 97% | 36.63% | Sensor histidine kinase |
| *Alistipes putredinis* | 445970 | B0MUZ2 | OKY96599.1 | 100% | 96% | Tryptophanase |
| *Sphingobacterium paucimobilis* | 1346330 | U2J6M1 | WP_021069213.1 | 100% | 100% | DoxX family, membrane protein YphA |
| *Clostridium hylemonae* | 553973 | C0C5Y6 | WP_006444869.1 | 100% | 100% | Sugar ABC transporter protein |

Furthermore, TnaA (encoding indole) was previously reported in *E. coli*[50] and *Enterobacteriaceae*[51], which is also indicated by the QSHGM, suggesting that there will be indole-based interaction between these two microbes. Therefore, a microbial consortium including *E. coli* O157:H7, *B. pectinophilus* ATCC 43243 and *E. bacterium* 9_2_54FAA can be regulated by manipulating the concentration level of AI-2 and indole (Fig. 4b). Furthermore, QSHGM can enable the prediction of more sophisticated interaction networks. When introducing the *P. aeruginosa* PAO1 into the above three-strain consortium, there will be complex

microbial cell-cell communications based on AI-2, AHLs, and indole (Fig. 4c), in which the interactions between *P. aeruginosa* PAO1 and *E. coli* were reported and validated previously[52,53]. When adding *Burkholderia cepacia* GG4 to the above four-strain consortium, we can also predict the complex QS-based interaction network for a five-strain consortium that communicates with AI-2, indole, AHLs, HAQs, and DSFs (Fig. 4d), which included a previously validated HAQs-based interaction between *P. aeruginosa* and *B. cepacia* GG4[54]. To sum up, QS-based interaction predictions stated above have been partially verified in the

**Table 2 Results of another eight expanded entries without existing annotations based on Phyre[2].**

| Strains | TaxID | Entry | Template | Confidence | Coverage | Annotations |
|---|---|---|---|---|---|---|
| *Bacillus amyloliquefaciens* | 1390 | A0A5C8IUS9 | c5xybB | 100% | 97% | AimR transcriptional regulator |
| *Bacillus mycoides* | 1405 | A0A1W6AJT8 | c5zvvA | 100% | 90% | AimR transcriptional regulator |
| *Bacillus thuringiensis* | 56955 | A0A243M9P9 | c5zw5A | 100% | 95% | AimR transcriptional regulator |
| *Bacillus amyloliquefaciens* | 1390 | A0A5C8IY56 | c5zvvA | 100% | 99% | AimR transcriptional regulator |
| *Bacillus atrophaeus* | 720555 | A0A0H3E1W6 | c5zvvA | 99.90% | 98% | AimR transcriptional regulator |
| *Bacillus atrophaeus* | 720555 | A0A0H3E2G4 | c5zw5A | 100% | 100% | AimR transcriptional regulator |
| *Lysinibacillus fusiformis* | 28031 | A0A4Y4IIW5 | c6mfvC | 100% | 90% | Signaling protein (tetratricopeptide repeat) |
| *Streptococcus salivarius* | 1304 | A0A5C4P2T9 | c4bxiA | 99.90% | 33% | ATP binding domain of AgrC |

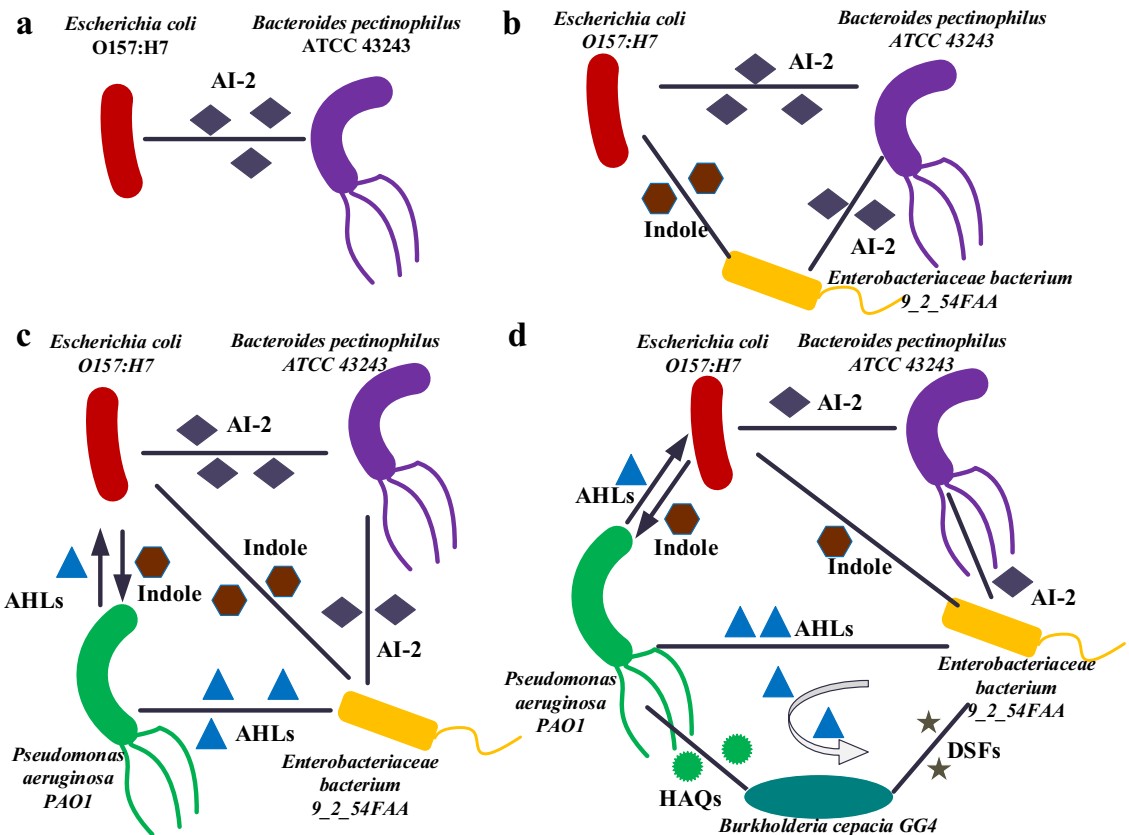

**Fig. 4 QS-based communication predictions for various microbial consortium. a** Two-strain communication based on AI-2; **b** three-strain communication based on AI-2 and indole; **c** four-strain communication based on AI-2, indole, and AHLs; **d** five-strain communication based on diverse QS languages.

corresponding experiments from other reported researches. Therefore, it has huge potential to predict more complex QS-based interaction networks including multi-component strains based on diverse QS languages.

**QS communication network construction.** Microbes communicate via various QS signals (also termed as microbial languages), and it is possible to construct a cell-cell communication network among different gut microbes based on diverse QS languages, which we termed as "QS communication network" (QSCN). Based on a review of previous studies (Supplementary Table 2), we decided to focus on the common nine QS languages, i.e., AHLs, DSFs, HAQs, CAI-1, AIPs, Dialkylresorcinols, Photopyrones, indole, and AI-2 to construct the proposed QSCN. With the help of the QSHGM and several hypotheses (details given in Supplementary Table 3), we firstly constructed an undirected QSCN for the 818 gut microbes based on the "speaking" of the above nine QS languages (Fig. 5a) (Supplementary Data 13). This intricate network visualizes complex QS-based communications among human

gut microbiota. Different microbes are linked together through various languages to form a microbial communication network, and connections could be used to regulate microbial interactions between themselves and the surrounding ones. Most of strains produce AI-2 (567, 69.3% of 818 gut microbes) as the communication language, followed by HAQs (332, 40.6%), DSFs (325, 39.7%), CAI-1 (259, 31.7%), Dialkylresorcinols (129, 15.8%), Photopyrones (107, 13.1%), indole (77, 9.4%), AHLs (64, 7.7%), and AIPs (22, 2.7%).

Note that multiple microbes can speak one common language which is in line with the interspecies crosstalk[55]. Taking six typical languages (AHLs, CAI-1, HAQs, DSFs, Indole, and AI-2) as examples, we found that there are 64, 40, 22, and 5 species sharing two, three, four, and five QS languages, respectively (Fig. 5b). AI-2 also ranks first with the highest genus-level counts (138 genus) than the other languages, which is in line with what has been broadly observed[13]. Many overlaps of AI-2 or indole being spoken among different microbes, which also indicates that both of them are widely recognizable languages playing a major

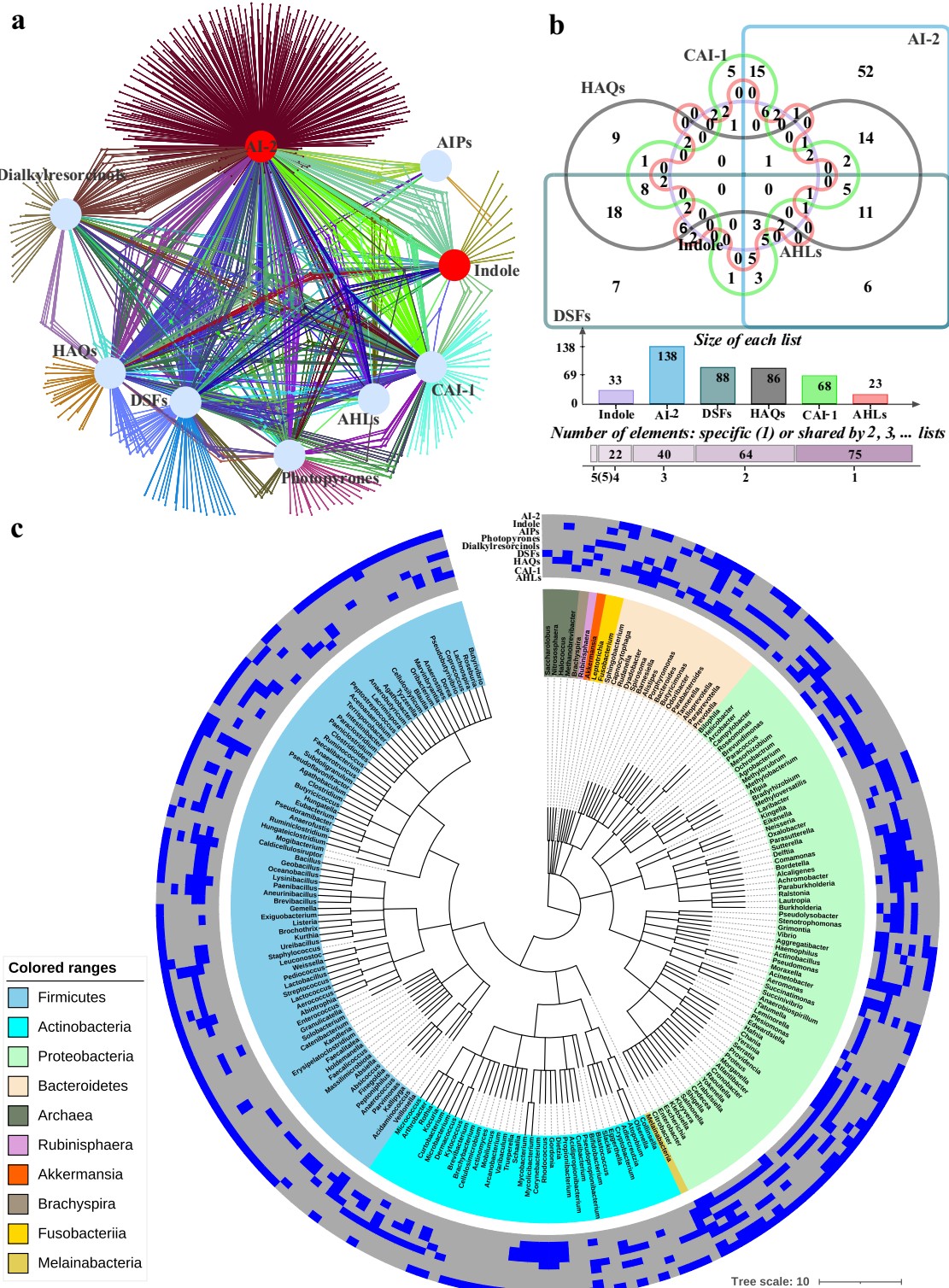

**Fig. 5 QSCN for human gut microbiota based on diverse QS languages. a** QSCN for 818 human gut microbes based on nine languages (AHLs, DSFs, HAQs, CAI-1, AIPs, Dialkylresorcinols, Photopyrones, indole, and AI-2). Generally recognized intraspecies and interspecies languages are marked in blue and red, respectively. Note that the network diagram was generated using EVenn[64]. **b** Microbial genus distribution for six typical QS languages, i.e., AHLs, CAI-1, HAQs, DSFs, Indole, and AI-2. **c** Hierarchical clustering of nine QS languages found in 210 human gut microbial genus. The constructions are classified into ten genus-level clusters based on their phyla and taxonomy. Microbial genus from Firmicutes is colored in blue; Actinobacteria, cyan; Proteobacteria, green; Bacteroidetes, yellowish. Heatmap on the outermost layer indicates QS languages distribution in each cluster, existence is colored in blue; no existence, gray. Source data are provided as a Source Data file.

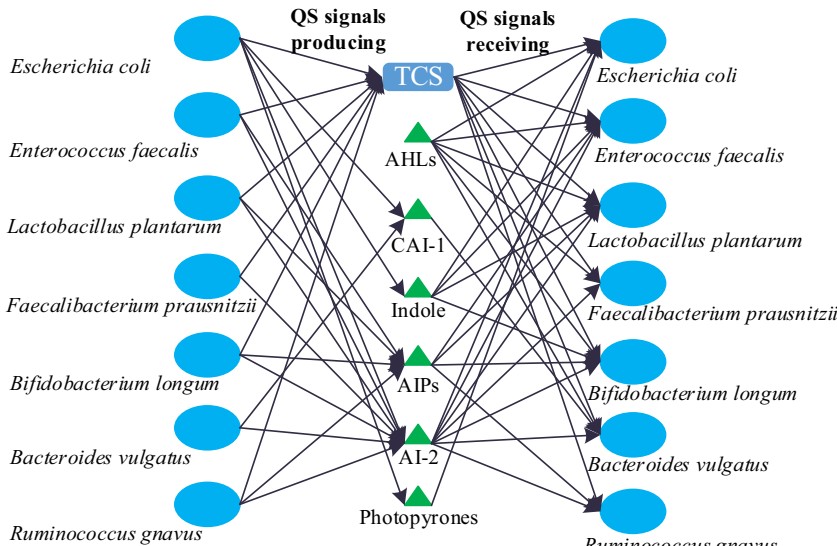

**Fig. 6 Typical small QSCN that includes QS signals producing and receiving for seven human microbes.** Note that the seven-strain simplified human microbiomes are taken from Colosimo et al.[42].

role in interspecies communications[56,57]. We found that those traditionally often considered intraspecies languages (AHLs, CAI-1, HAQs, and DSFs) may also be involved in some interspecies communications. Like Scott et al.[58], we also realized that the crosstalk of different QS languages implies the redundancy of microbial languages that is potentially helpful for the stability of natural microbial systems.

The QSCN was constructed based on the 818 human gut microbes, which include mainly Firmicutes (79), Actinobacteria (36), Proteobacteria (69), Bacteroidetes (16), and others (10). We have collected and sorted the nine QS languages for 210 microbes at the genus level, shown by the heatmap representation in Fig. 5c to gain a better understanding of the QSCN (Fig. 5a). As in previous studies, we also found that AHLs exist only in Proteobacteria[59], AIPs exist mostly in Firmicutes[12], and other QS languages are distributed in-homogeneously in the whole genus-level microbes[60]. Surprisingly, there is no highly similar distribution of QS languages within the same genus-level microbes. For example, the distribution of QS languages in Actinobacteria is quite different (Fig. 6c, cyan). This suggests that the existence and evolution of QS synthases in microbes might have not been strictly conserved at the genus level, but are more likely to be related to some other factors, such as environmental factors and spatial distributions[61–63]. To sum up, the distribution of QS languages suggests the diversity of the microbial languages, the complexity of cell-cell communication, and the redundancy of QS-based interactions among human gut microbiota.

The QSCN we presented above (Fig. 5a) is an undirected and bipartite network involving two types of nodes, namely QS languages and microbes. This network can be projected to a one-mode network that visualizes microbial communication-based interactions directly. The giant network would consist of 801 nodes connected via 190,580 edges (Supplementary Fig 4). The largest degree in the giant network is 771, while its average degree is 237.93. The dense QS network is similar to other microbial interaction networks that carry high degrees for individual strains[65,66]. Key nodes in this network were selected from 5% of the total nodes (40 nodes, Supplementary Table 4) of the network with a large degree and high betweenness centrality[67]. Note that all the 40 key nodes are Firmicutes, Bacteroidetes, or Proteobacteria, (Supplementary Table 4) which are known to be dominating species of the human gut microbiota[68,69]. Therefore,

QSCN can be projected to a one-mode network and shrunk further to be the complete graph with 40 core gut microbes (Supplementary Fig. 5). While such a dense network more likely approximates a theoretical maximum set of QS-based interactions it nevertheless indicates excellent microbial communications among the core microbes. As such, what is visualized here is essentially a sub-network with a particularly high "density", not representative of the entire network of the whole gut environment. We would also like to point out that, although each microbe can produce so many QS languages in the 40 core gut microbes, the specific intensity for each language cannot be provided in this work; the intention is that we determine the existence of the QS-based communications (as done in this work), and then to investigate its corresponding intensity (future work), eventually bringing a comprehensive understanding of the communication-based microbial interactions.

**Discussion**

This work has been based on several hypotheses on microbial composition, language types, TCS function, non-cheating ecology, and QS crosstalk, which may be addressed in the future to improve the accuracy and completeness of the QSCN (Supplementary Table 3). Besides, the large number of links in our QSHGM Database and the QSCN means that it is inevitable that there will be some false positive relations. Even if there is no problem at the level of individual nodes, the relations we have predicted were not necessarily always true in reality. Note that many TCS entries possess QS functionality (Supplementary Data 1, Supplementary Fig. 8), but not all of them would do so, which would apply to a portion of the TCS entries collected into our Dataset III that was built with the intention of collecting as many potentially QS-relevant entries as we can, let alone these entries would still be relevant to inter-cellular communication. To mine more potential QS entries, we combined manual curations, BLASTP-based expansion, and ML-based classifications together in this work along with minimizing false positives as possible. On the other hand, QS links we predicted based on the database would be "possibilities", not reality, and still require experimental verification. We offered a tool to allow users with various applications in mind to see the "possibilities" in the first place, which allows them to subsequently focus their experimental verification.

Note that short peptides (such as AIPs) and proteins are not generally placed together for sequence BLASTP and functional analysis, because proteins generally have a fixed structure while short peptides do not. AIPs sequences can also easily lead to the increasing of false positives from the BLASTP process. The four ML-based classifiers were trained on AA frequencies, which were not accurate for the prediction of the short sequences such as the AIPs (about 5–30 amino acids), of which the physicochemical properties[70], the information on amino acids combinations with fixed length[43], and even the composition of common amino acids were not complete. Therefore, to increase the reliability of the expansion, we have removed the signal peptides in the BLASTP-related datasets (I and VII), thus leading to sparse edges for the "AIPs" node in our QSCN (Fig. 5a). This calls for a more accurate method to cover more aforementioned amino acid features for short sequences to mine the potential signal peptides in the future to make the QSCN more complete. Furthermore, the nine QS languages studied in this work (Supplementary Table 2) did not include all existing QS signals, such as Autoinducer-3 (AI-3, with unclear synthase sequence)[71], let alone new QS languages that would be discovered in the future. Considering the QS crosstalk widely exists in nature[55], we also hypothesized that microbes that speak the same type of languages (such as AHLs) can communicate with each other. Future works should be conducted to quantify the specific intensity of diverse QS crosstalk for the same type of languages, such as the AHLs with different side chains. Therefore, more AIPs, some novel QS entries, and their corresponding weighted networks of different QS languages for more gut microbes will gradually be updated in our QSHGM and QSCN.

Bipartite (Fig. 5a) or one-mode QSCN (Supplementary Fig. 4) illustrates diverse language connections, which however lacks the further interactions between QS languages senders and receivers. By differentiating QS signals producing and receiving with the help of both QS synthases and receptors, there is potential to construct a directed and more precise QSCN. Taking the seven-strain simplified human microbiomes from Colosimo et al[42] as an example, we constructed a typical small precise QSCN that includes QS languages producing and receiving (Fig. 6). QS language receiving (Fig. 6, right) is more complicated than language producing (Fig. 6, left), which indicates that some microbes can receive a particular QS signal without producing it. This phenomenon is consistent with the previously observed QS cheating behavior in certain microbes, such as *P. aeruginosa*[72] and *E. coli*[73]. However, the reliable construction of the directed and precise QS networks still faces many challenges, such as the huge network scale, multi-layer control structures, complex QS crosstalk, intricate social cheating, diverse environmental factors, and different spatial distributions, and insufficient QS entries for many uncultured microbes. Nevertheless, we expect that the further directed and precise QSCN including QS languages producing and receiving will receive increasing attention from future research which will be engaged in developing more knowledge and technologies for various gut microbes, aiming to construct the valuable precise QSCN which can be regarded as one of the key knowledge maps of the human gut system.

Microbial communities and their functions are shaped by both metabolic interactions and communication-based regulations. Microbe–microbe interactions based on the exchange of metabolites have received much attention in microbial ecology[65,74]. At the same time, various two-strain or three-strain synthetic consortia have been constructed by implementing QS for stabilizing the microbial ecosystem (more details in Supplementary Table 5). As we proposed earlier, QS-based communication networks (QSCNs) can be vertically and horizontally applied to the regulations in natural microbial systems and synthetic microbial

consortia, respectively[24], and they play different roles than metabolic integration networks (MINs) (more details in Supplementary Table 6). On the other hand, a QSCN and a MIN can be co-present and function collectively in a microbial ecosystem (Supplementary Fig. 6). One such example is from our earlier work[75], where we developed combinational QS devices for automatic dynamic control in a cross-feeding cocultivation of a synthetic community, to achieve the optimization of the system which simultaneously involved QS communication, cell growth competition, and cooperative production. More recently, a methodology was proposed for designing robust synthetic communities that include competition for nutrients, and use QS to control amensal bacteriocin interactions[76], which can be considered as a more generalized example of how the combination of QSCNs and MINs could lead to desirable designs of engineered microbial consortia.

To illustrate the potential of complementary use of the QSCN constructed in this work for the gut community and a MIN, here we consider the work of Venturelli et al[65] on a simplified human microbiota consortium (SIHUMI). By comparing the inferred total interaction network (Supplementary Fig. 7a) with the MIN (Supplementary Fig. 7b), it was recognized that the former was significantly denser than the latter with several prominent inter-microbial links not associated with the exchange of metabolites, which were considered to be possibly mediated by signaling molecules instead[65]. Applying our QSHGM database to the SIHUMI community, we have obtained a bipartite QSCN (Supplementary Fig. 7c), which shows specific QS-based communications that offer plausible mechanisms for links that the MIN could not explain (Supplementary information, Section 5). Thus, we consider our QSHGM database as a tool that can facilitate the identification of possible QS-based inter-microbial interactions which may complement metabolic exchanges in a complex community in explaining an observed community structure; such possible interactions can be tested based on the microbe-QS signal pairs suggested by the database through e.g. detecting and manipulating the excretion/reception of the specific QS signaling molecules involved.

Various QS-based interactions play an essential role in the regulation of homeostatic states, metabolism, and immune responses in the human gut system. Therefore, constructing a comprehensive QS database for the human gut microbiota is highly desirable for making gut microbiology more predictable and for developing potential therapies for diverse gut diseases. In this work, we developed a systematic workflow including collecting, expanding, and mining modules to construct a comprehensive QS repository for the human gut microbiota. Machine learning algorithms including SVM, RF, KNN, and DNN were combined with protein annotations, functional analysis, and homologous modeling to facilitate the efficiency of data collection and mining. As a result, we established the QSHGM (http://www.qshgm.lbci.net/, with browsing and searching functions) which contains 28,567 redundancy removal entries for 818 human gut microbes.

With the help of the QSHGM, users can search many QS-based interactions for various microbial consortia based on diverse QS languages. We constructed a QSCN to visualize and decipher intricate QS-based interactions for human gut microbiota. We found that the distribution of QS languages in microbes is not strictly conserved at the genus level, but is more likely to be related to other factors. There are significant genus-level overlaps between microbes on what are commonly regarded as intraspecies languages, which suggest that these languages may also be involved in some interspecies communications. The predicted sharing of various subsets of the QS languages between microbes supports the notions of the diversity of microbial language and

the redundancy of cell-cell communications, which are helpful for maintaining the stability of natural microbial systems. The QSCN can be projected to a one-mode network; a fraction of which is a sub-network representing potentially very "dense" communications of 40 core gut microbes. This work contributes to the construction of the QSCN for human gut microbiota that may form one of the key knowledge maps of the human gut system in the future. Such a network holds huge potential for improving our understanding of the dynamics and resilience of gut micro-biology and for developing applications such as potential thera-pies. For the QSCN to be more effective and more widely applicable, further work is needed to identify the strengths of diverse QS-based interactions and combine it with other types of connections, particularly those captured by microbial interaction networks, to achieve reliable, quantitative predictions for micro-bial ecosystems.

QSHGM and QSCN can not only give us a better under-standing of QS-based microbial communication principles but also will do much help in providing new manipulations to syn-thetic microbiota and developing potential therapies (Supple-mentary Fig 9). Thanks to the large scale of the data established in this work, potential useful details for the QS-based communica-tions among different gut microbes can be obtained in our QSHGM database. At the strain level, QSHGM and QSCN will provide user-friendly data searching, and assist the scientific community in various interferences and manipulations of QS systems to alleviate antimicrobial resistance, inhibit pathogenic bacteria, and develop new QS-based synthetic gene circuits for various applications. At the community level, the communication-based regulations can be visualized for human gut microbiota, users can search many QS-based communications for various microbial consortia including multi-component strains based on diverse QS languages. The predicted commu-nications will provide guidance for consortia-based therapies or constructing new synthetic microbial consortia. Furthermore, QSHGM furnishes high-throughput data for large-scale QS-relevant statistical analysis.

## Methods
**Data acquisition**. QS is a common mechanism which includes autoinducer synthase and relevant QS receptors[77]. For most Gram-negative bacteria, the autoinducer produced by the autoinducer synthase accumulates in the culture with the cell density increasing; When the concentration of the autoinducer reaches a certain threshold, it will diffuse back into strain and be recognized and bonded by the QS receptor to be a complex to activate or inhibit the transcription of downstream genes[56]. The auto-inducer synthases and receptors for Gram-negative bacteria from SigMol (http://bioinfo.imtech.res.in/manojk/sigmol), and QS receptors for Gram-positive bacteria from Quorumpeps (http://quorumpeps.ugent.be) are utilized as the validated QS proteins in our research. Their corresponding amino acid sequences are obtained from UniProt (https://www.uniprot.org/)[39]. Note that QS entries in Sigmol and QuorumPeps database without corresponding amino acid sequences were discarded in this study. We have also made some choices about the entries from Sigmol and QuorumPeps database to improve the efficiency of Local BLAST. For example, we kept only one related entry for the same QS entries, such as the S-ribosylhomocysteine lyase (LuxS), Acyl-homoserine-lactone synthase LuxM, and accessory gene regulator C (AgrC). The autoinducer peptides (AIPs), such as Nisin precursor peptide (NisA) and competence stimulating peptide AgrD, were not considered in the BLASTP-related work. Because the signal peptide sequence of Gram-positive bacteria is rela-tively short (about 10–30 amino acids), which will be easy to increase the false positives for the BLASTP. Therefore, to increase the reliability of the ML-based classifiers, we have improved the signal peptides in the BLASTP-related datasets (I and VII) by more exact matches (with lower E-value) and manual checking of their protein annotations or the open reading frames (ORF) in the NCBI database. In addition, 818 gut microbes from a virtual metabolic human database (www.vmh.life)[28] are regarded as the human gut microbiota in this study, and their corre-sponding proteomes are also obtained from UniProt.

**Feature extraction and classifiers development**. The secondary and tertiary structure of a protein depends on its amino acid sequence[78]. In this study, the information of amino acids in protein sequences was calculated with the help of the iFeature package, ML algorithms (SVM[79], RF[80], and KNN[81]), and deep learning

algorithms (DNN[82]) were trained on the carefully curated positive and negative samples to develop different classifiers. We calculated the frequency of each amino acid type in each QS-related protein sequence. The frequencies of all 20 natural amino acids are the percent of the number of amino acid types divided by the length of a protein sequence[43], which is listed as follows:

$$f(t) = N(t)/N, t \in \{A, C, D, \dots, Y\} \quad (1)$$

where $N(t)$ is the number of amino acid type $t$, while $N$ is the length of a protein or peptide sequence.

**Positive and negative samples construction**. With the help of the evolution analysis of amino acid sequences of autoinducer synthases and receptors, we collected the reported and annotated QS proteins for 818 gut microbes as positive samples. The finally obtained positive samples (Dataset III) were the arrays of 21,383 amino acid frequencies of the collected QS entries. In the collecting module, we did an evolu-tionary analysis for the validated QS entries to propose a possible cluster rule for negative samples collection with the help of MEGA X[83] and iTOL[84]. The evolutionary history was inferred using the Neighbor-Joining method[85]. The tree is drawn to scale, with branch lengths in the same units as those of the evolutionary distances used to infer the phylogenetic tree[86]. The evolutionary distances were computed using the Poisson correction method and are in the units of the number of amino acid sub-stitutions per site[83]. We constructed negative samples by removing QS-related components from typical Gram-negative bacteria (*Aliivibrio fischeri*, *Escherichia coli*, *Pseudomonas aeruginosa*, *Salmonella typhimurium*, and *Vibrio parahaemolyticus*) and Gram-positive bacteria (*Bacillus subtilis*, *Staphylococcus aureus*, and *Lactococcus lac-tis*), and removing proteins that directly and indirectly associated with QS, i.e., cluster rules, such as quorum sensing, luxR, two-component, homoserine-lactone synthase, histidine kinase, biofilm, autoinducer, bacteriocin, competence, virulence, signal, sensor, response, regulator, membrane, binding, transcriptional activator, etc. Sub-sequently, we got an output array of 22,780 (negative Dataset IV) amino acid fre-quencies, which were calculated from amino acids sequences of proteins of the above eight organisms after removing QS-related entries.

**ML-based classifiers**. The amino acid composition (AAC) calculates the fre-quency of each amino acid type in a protein or peptide sequence. We calculated AAC in each entry sequence as the protein features. "model.py" was created for training samples with SVM, KNN, and RF (random forest), "nn.py" was the script used for training samples with Neural Network. Classifiers were trained and validated based on the positive and negative samples, and then tested on dataset VII (Fig. 2). Performances of the four ML-based classifiers were measured based on the accuracy, precision, recall, and F1 score, which are defined as follows[87].

$$\text{Accuracy} = (TN + TP)/(TN + FP + FN + TP) \quad (2)$$

$$\text{Precision} = TP/(TP + FP) \quad (3)$$

$$\text{Recall} = TP/(TP + FN) \quad (4)$$

$$F1 = 2 * \text{Precision} * \text{Recall}/(\text{Precision} + \text{Recall}) \quad (5)$$

where TP represents true positives, TN denotes true negatives, and FP and FN are false positives and false negatives, respectively. F1 score is the harmonic mean of prediction and recall. The higher the F1 score is, the better performance the classifier will be of.

All the four classifiers were applied to predict whether the input amino acid sequences are QS entries or not with the output being 1 (yes) or 0 (no), respectively. SVM is a commonly used supervised ML algorithm in protein prediction[87]. The basic idea of SVM is to find the separated hyperplane in a very high-dimension feature space that can correctly partition the training dataset[79]. SVM can also integrate kernel functions, which makes it to be a nonlinear classifier. In this study, for our results, we applied the radial basis function (RBF) with standard deviation $\sigma = 0.125$ and set regularization parameter $C = 4$ to train the positive and negative samples. The GridSearchCv code[88] was used to select and determine the optimal combination of hyper-parameters automatically to achieve the best performance.

K-nearest-neighbor (KNN) is also a traditional classification method when there is little or no prior knowledge about the distribution of the data[89]. The principle behind KNN is to find $k$ training positive and negative samples nearest in the distance to the new point and predict the label from these samples. Firstly, the distance between the test sample point and each other sample point is calculated, then each distance will be sorted and $k$ points with the smallest distance will be selected, and the categories of K points will be compared and classified. We used a MultiScheme package in WEKA to choose between 12 KNN models (1, 3, 5, 10, 20, 30, 50, 100, 150, 200, 250, and 300) and the KNN with $k = 5$ yielded the best result.

Random forest (RF) is a classification algorithm that uses a set of decision trees[80]. Each decision tree is constructed by using a sample of training data, and each segmentation candidate set is a subset of random characteristics. RF has been proven to have excellent performance in classification tasks[82]. In this study, positive and negative samples are randomly selected from the original data to construct the sub-training set to generate the decision tree. At each node, we

**11**

randomly selected the $n$ child variables ($n \ll N$) from the $N$ input variables. The optimal segmentation coefficients on these $N$ sub-variables are used to segment the nodes. The $n$ value remains constant during the growth of the forest. For new samples, the classification results can be obtained by voting on these decision trees. $N$ is generally taken as the square root of the dimension of the eigenvector of the input samples. Here, we set $n\_estimators = 122$ (the number of trees in the forest), and $max\_depth = 55$ (maximum depth of the tree). Other hyper-parameters were also generated and selected with the help of GridSearchCv[88].

Neural networks (NN) play an essential role in biomedicine[82], antiviral peptides prediction, protein–RNA interaction[90], and protein data mining. For regular neural networks, the most common layer type is the fully-connected layer in which neurons between two adjacent layers are fully pairwise connected. In the input layer, there are a certain number of neurons corresponding to input features. In the first layer (one-to-one layer), the same number of neurons are used, and each is connected to one neuron from the input layer. Then we added two hidden layers after the one-to-one layer. The first hidden layer is fully connected with the one-to-one layer and the second hidden layer is fully connected with the first hidden layer. The last layer is an output layer which only has two neurons. Batch normalization was applied to a one-to-one layer and each hidden layer to accelerate the training process. SGD optimizer was used to train the DNN model and the learning rate was fixed as 0.01. Default values of the other hyper-parameters of the DNN model were set to default ones without tuning.

**Reporting summary**. Further information on research design is available in the Nature Research Reporting Summary linked to this article.

## Data availability

About 28,567 redundancy removal entries for 818 gut microbes generated in this study have been deposited in our QSHGM database, which is freely available at: (http://www.qshgm.lbci.net/). We will continuously update the database QSHGM. Computer-readable tables generated in this study are provided in Supplementary Data 1–13. More details for the relevant data of QS entries from Gram-negative microbes, QS entries from Gram-positive microbes, 818 human gut microbes, and their corresponding proteomes can be searched in SigMol[29], Quorumpeps[30], VMH[28], and UniProt[39] databases, respectively.

## Code availability

We also used Python 3.7 to write the method and analyse the collected data to construct ensemble classifiers. The codes have been provided in a GitHub repository at: https://github.com/guofei-tju/qshgm-code[91].

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

## Acknowledgements

This study was supported by the National Key Research and Development Project of China (No.2019YFA0905600, J.Q.), the National Natural Science Foundation of China (62172296, F.G.), the Funds for Creative Research Groups of China (21621004, J.Q.), and National Key Research and Development Program of China (No. 2020YFA0907900, J.Q.).

## Author contributions

J.Q. conceived the project. F.G. and A.Y. designed the project. S.W. conducted the systematic workflow and relevant analytical calculations. J.F. trained the reported data with different ML algorithms and constructed the database. H.W., Z.Q., J.G., S.S., X.H., Y.L., and X.W. collected the reported and annotated QS entries. All authors analysed the results. S.W. wrote the manuscript. C.L., A.Y., F.G., and J.Q. edited the manuscript.

## Competing interests

The authors declare no competing interests.
