## [Peer Review File · Nature Communications]

Reviewers' Comments:

Reviewer #1:

Remarks to the Author:

The article "Machine Learning Aided Construction of the Quorum Sensing Communication Network for Human Gut Microbiota" by Wu et al. describes the construction of a new biological database, QSHGM, that contains information on the quorum sensing (QS) repertoire of 818 human gut microbial species. The database was constructed by a computational workflow that uses verified QS-related amino acid sequences, genome annotations, and a machine learning workflow to mine proteomes for thus far missed QS traits. The database is then used to construct potential communication networks between gut microbial species. The work is important and the way it was conducted appears appropriate, but it suffers from a lack of clarity which needs to be improved upon. The work is of interest to the microbiome field, but several major points ought to be addressed before publication.

Major comments:

- The language throughout must be improved, e.g. L22 (limited QS entries?), L181: What does overlap relate to here? L211ff- sentence unclear.

- Why was "two-component" used as a QS search term? This should yield numerous false positives.

- Data provision & clarity:

While the authors provide a web interface (which is nice), the intermediate datasets must be provided in a computer readable format for reproducibility, especially Dataset I, IV, III. In particular,

o please provide dataset 1 obtained from Sigmol and QuorumPeps; what are the 213 QS entries (and why is this number different from the supplementary table of the cited resource by Bernard et al, 2020?

o Why are there some amino acid sequences in the "positive samples" that are annotated as irrelevant for QS (why are they "positive"), e.g. Figure 3B?

o Figure 1 needs numbers of sequences for each of the data sets.

o Please provide the final QSHGM as a csv/machine readable file (the web interface is nice for browsing, but without an API it would be good to share the final data in a way that facilitates future work).

- Why did you only use receptors for Gram positive, but synthases and receptors for gram negatives?

- Machine learning: did you truncate / normalize amino acid sequences (features) to a consistent length to predict labels (negative/positive)? The text needs some improvements to clarify what was done. What exactly were features? Why did you choose multiple classifiers, all of which seem to perform similarly? How did you perform ensemble predictions (majority across the classifiers)? Can you comment on the potential that there different numbers of positive samples from Gram positive vs negative QS systems, i.e. can that confound downstream analyses?

- Please justify and explain better what negative samples are. Are they aminoacids that do not have a relationship with QS, sampled from the proteomes of the named organisms? Why was this approach chosen to build a training set (as opposed to perhaps selecting from other secretion- but not QS related sequences)? I do not think anything is wrong here, necessarily, but a better explanation would help.

- Figure 2 (panel F in particular) needs much more annotation and explanation in the caption.

- Explain better what the communication networks are showing. E.g. Figure 5A) Does this mean both *E. coli* and *Bacteroides pectinophilus* could be 'listening to' and 'speaking' the AI-2 language? Figure 5D shows arrows and lines, and a thick white arrow, what do they mean? Please improve the clarity of the figure and annotate all that is shown in the figure or the caption.

- Could you provide a computer readable table for the calculated communication networks?

Minor comments:

L53: re: AHL signal incorporation into plasma membranes — the reference is talking about lung epithelia. This seems not a relevant point unless also true in the gut.

L388: Doubtful that evolution works for the good of the population.

L361 and 439: I was confused by the use of the word "familial" and genus in the same sentence

Reviewer #2:

Remarks to the Author:

In the present work, the authors built a workflow for QS entries collecting, expanding and mining. All these QS entries are proteins. Then four ML-based models were trained to check if a protein is a QS entry. There are 21,410 positive samples were collected manually and 7,157 QS entries were predicted by ML-based models, and the total 28,567 QS entries contain 1,882 QS synthases and 26,685 receptors. Based on QS entries and QS-based microbial interactions, the authors developed the QSHGM database. Finally, a network was used to visualize the relationships among microbes based on the QSHGM database.

My major concern is how this QS-based communication network help improves our understanding of the dynamics and organizational principles of microbial ecosystems, which originally is considered largely based on metabolic interactions. The authors just briefly mentioned this in the discussion part. But a quantitative model/analysis (as an example) is needed, which is quite important.

Other comments/questions are as follow:

1. About the workflow

1.1 The schematic diagram of the workflow with too many different datasets, styles, colours, and unclear description is very hard to follow. The input (data source), output and filtering rules should be defined clearer for each step. A table would be quite helpful.

2. ML-based models

2.1 The authors used 4 different models: random forest (RF), KNN, SVM, and DNN. The authors claimed that "the RF classifier achieves the best performance" based on accuracy and F1 score (line 204). Why did the authors use all four models instead of the best model?

2.2 The authors mentioned "four ensemble classifiers" (line 126). How did the authors combine 4 different algorithms together by ensemble learning?

2.3 There are 21,410 positive samples (line 248), but the number of total samples (proteins) in 818 gut microbes and the number of negative samples is unclear. Is a sample either positive sample or negative sample?

2.4 The authors mentioned that 21,410 positive samples were collected manually. The search was based on four commonly used QS annotations, i.e., "quorum sensing", "LuxR", "two-component", and "tryptophanase" (line 131). Do all "two-component regulatory systems" work as quorum sensing? Are these four keywords specific enough to narrow the searching scope and reduce the false-positive QS entry?

2.5 The authors mentioned that "All the four classifiers were applied to predict that whether the input amino acid sequences are QS entries or not with the output being 1 or 0, respectively" (line 488). However, the input for each algorithm, which is how to represent an amino acid sequence

with variable length, was not mentioned in the draft.

2.6 There are two different QS entry types: QS synthase and receptor, and the total 28,567 QS entries contain 1,882 QS synthases and 26,685 receptors. Receptors must also dominate the training set. Is there any bias when we predict QS synthases since the sequences of QS synthases may be different from receptors' dramatically?

3. QS-based microbial interactions

3.1 The authors claimed that "according to the collected data in the QSHGM database, we can predict various potential pairwise QS-based microbial interactions" (line 289) and "predict more sophisticated interaction networks" (line 298). However, the details of this prediction method cannot be found in this draft. Did the relation between two microbes include in the database? If so, the data source should be pointed out.

3.2 According to Fig. 7B, one microbe (such as *E. coli*) can produce compounds (such as CAI-1) by QS synthases, and another microbe (such as *Bacteroides vulgatus*) can receive the same compound by its receptor, so they can communicate with each other. Since each microbe can produce so many compounds and each microbe also can receive so many different compounds, Fig. 7A gives a complete graph that links every node to every other node for the 40 core microbes. We cannot figure out that what information this graph can provide.

3.3 Same concern as 2.4 and described in 3.2, there are so many links between each different species. The false-positive relations between each different species should be tackled carefully.

Response to Reviewer #1 (Expertise: microbiome, computational biology, Machine learning):

The article “Machine Learning Aided Construction of the Quorum Sensing Communication Network for Human Gut Microbiota” by Wu et al. describes the construction of a new biological database, QSHGM, that contains information on the quorum sensing (QS) repertoire of 818 human gut microbial species. The database was constructed by a computational workflow that uses verified QS-related amino acid sequences, genome annotations, and a machine learning workflow to mine proteomes for thus far missed QS traits. The database is then used to construct potential communication networks between gut microbial species. The work is important and the way it was conducted appears appropriate, but it suffers from a lack of clarity which needs to be improved upon. The work is of interest to the microbiome field, but several major points ought to be addressed before publication.

Response:

Thanks for the positive comments. The point-to-point response to individual comments is listed as follows.

Major comment

Major comment 1: The language throughout must be improved, e.g. L22 (limited QS entries?), L181: What does overlap relate to here? L211ff- sentence unclear.

Response:

Sorry for these ambiguities. The language has been re-edited carefully.

“limited QS entries” has been re-edited as “a small number of QS entries” in the revised manuscript.

“Overlap” means that one protein belongs to different protein clusters. For example, P69409, P0ACZ6, P0AGA8, P66798, P0AF30, P0AEL9, and Q8XE66 in the *E. coli* O157:H7 are both LuxR-type and TCS receptors (Fig. 2E).

We have re-edited this part in the revised manuscript. Hope the revised version is easier to understand.

Major comment 2: Why was “two-component” used as a QS search term? This should yield numerous false positives.

Response:

We recognize the possible confusion around the two concepts of QS and Two-component signal transduction systems (TCSs) in this work and would like to clarify this important issue. TCSs play an important role in microbial communications,

which have a certain overlap with QS [2], but it is difficult to separate the two clearly. Generally, various QS systems can be roughly divided into three types: (i) acylated homoserine lactones (AHLs) and other autoinducers received by LuxR-type receptors utilized by Gram-negative bacteria; (ii) auto-inducing peptides (AIPs) and other autoinducers sensed by two-component systems utilized by Gram-positive bacteria; and (iii) autoinducer 2 (AI-2) and indole for interspecies communication of microbial communities [3]. Here, we have listed functions of the 213 QS entries collected into Dataset I (more details in Supplementary Table 1) in Figure R3 (Figure S6) to indicate the overlap of QS and TCS, which shows that TCSs form an important part of QS entries. Therefore, the search for positive samples was based on three commonly used QS (“quorum sensing”, “LuxR”, “tryptophanase (indole synthase)”) and one TCS (“two-component”) annotations to collect the reported QS&TCS entries.

Figure R3. Function distribution for the 213 collected QS entries.

While there is strong evidence from the 213 entries mentioned above that many TCS entries possess QS functionality, we agree that not all of them would do so, which would apply to a portion of the TCS entries collected into our Dataset III which was built with the intention of collecting as many potentially QS-relevant entries as we can. On the other hand, we would like to point out that the entries in Dataset III (QS&TCS) was subsequently used as positive samples to train the classifiers; the predicted positive entries by applying the trained classifiers to the Dataset VII (resulting from Local BLASTP) were eventually checked manually to confirm QS functions. Out of 9253 entries (Dataset VII), 7184 were confirmed to be QS relevant, which represents a very high percentage. Therefore, although it is difficult to say exactly what is the proportion of the TCS entries collected in our final database that are not QS relevant, we anticipate that the proportion is likely to be moderate. Nevertheless, there is a need to make an explicit statement about the existence of such proportion so that the users are warned of encountering none-QS TCS entries, even though these entries would still be relevant to inter-cellular communication.

Note that we have modified the corresponding descriptions in the revised manuscript and supplementary material to illustrate the above problem to avoid misunderstanding.

Major comment 3 (Data provision & clarity): While the authors provide a web interface (which is nice), the intermediate datasets must be provided in a computer readable format for reproducibility, especially Dataset I, IV, III.

Response:

Thanks for the good suggestion. In the revised supplementary material, we have added the summary table (Table R3, see also Table S2) and .xlsx files for all the datasets that appear in Figure 1 (Supplementary Tables 1-10).

Table R3. The details for the datasets listed in the Figure 1.

Dataset	Input	Output	Suppl. Table
I	Reported entries from Sigmol and Quorumpeps database	213 reported QS entries	1
II	Proteomes of human gut microbes from UniProt	Proteomes of 818 gut microbes from VMH.	https://pan.baidu.com/s/1o46nn1b7L5nvCqgpwW7Zlw Password: tfnx
III	Collected QS and TCS entries from Dataset II	Positive samples (21,383 entries)	2
IV	Remove QS and TCS entries in cluster rules from typical strains in Dataset II	Negative samples (22,780 entries)	3
V	Results of the Local BLASTP	Results of local BLASTP with $E \leq 10^{-5}$ (14,573 entries)	4
VI	Overlaps of entries in Dataset III and V	5,320 reported entries	5
VII	Entries by excluding Dataset VI for Dataset V	9,253 entries to be classified	6
VIII	Union of uncharacterized proteins from the positives of RF, SVM, KNN, or DNN classifiers	534 un-annotated entries to be mined	7
IX	The extended QS entries obtained by the union of four classifiers	7,184 extended entries	8
Output S3	Proteins without QS functions	438 false positives	9
X	The total entries from the reported and extended QS&TCS entries	28,567 redundancy removal entries	10

Major comment 4 (Data provision & clarity): please provide dataset 1 obtained from Sigmol and QuorumPeps; what are the 213 QS entries (and why is this number different from the supplementary table of the cited resource by Bernard et al, 2020?)

Response:

Thanks, we have provided the details for the 213 QS entries from Sigmol and QuorumPeps database in the Supplementary Table 1.

The reasons for the difference in the counts of QS entries between our study and the work from Bernard et al, 2020 [4] are explained below:

- (1) Due to the QS information required being different between the two studies, the number of QS entries we used in our work is different from that in Bernard et al, 2020. In our work, we not only need to collect the record of QS entries, but more importantly, we need to obtain the relevant amino acid sequences. Therefore, the QS entries in Sigmol and QuorumPeps database without corresponding amino acid sequences were firstly discarded in our dataset I.
- (2) Further choices about the entries from Sigmol and QuorumPeps database were made to improve the efficiency of the local BLASTP. For example, we kept only one related entry for the same QS entries, such as the S-ribosylhomocysteine lyase (LuxS), Acyl-homoserine-lactone synthase LuxM, and accessory gene regulator C (AgrC).
- (3) We have also excluded the autoinducer peptides (AIPs), such as Nisin precursor peptide (NisA) and competence stimulating peptide AgrD from our Dataset I to improve the accuracy of the local BLASTP. This is because the sequences of the communication peptides from Gram-positive strains are relatively short (about 10~30 amino acids), which could easily increase the false positives for the local BLASTP. We believe that reliable inclusion of the AIPs requires some other methods, which is the subject of our future work.

Note that Dataset I was used to blast with the Dataset II to expand the reported QS entries for human gut microbes, which is not the only way by which we have collected QS entries: A much larger set of reported QS entries was separately collected into Dataset III which includes entries curated in Sigmol, QuorumPeps database, the work from Bernard et al, 2020, and other QS entries.

Major comment 5 (Data provision & clarity): Why are there some amino acid sequences in the “positive samples” that are annotated as irrelevant for QS (why are they “positive”), e.g. Figure 3B?

Response:

We have revised the original Figure 1 and Figure 3B to make our workflow easier to

follow, which we believe will also resolve the issue raised here. As illustrated in the Mining module of the revised Figure 1, the expanded proteins from Dataset VII (9,253) were classified by the four ensemble classifiers, respectively. The union of the four positives were then divided into the annotated positives (AP) and uncharacterized positives (UP) (Dataset VIII). With the help of the function analysis (Supplementary Table 11), we re-annotated the 534 uncharacterized entries and mined new potential QS proteins. Furthermore, we (manually) conducted the function analysis and checked their specific annotations, sequence similarity, and domains for the annotated/re-annotated union of positives (Supplementary Table 11) to decide whether the entry has QS function (true positives) (Dataset IX, 7,184 entries, more details in Supplementary Table 8) or not (false positives) (Output S3, 438 entries, more details in Supplementary Table 9). *In other words, the some of the “positive entries” predicted by the classifiers were determined to be “false positive” by a subsequent step of manual checking.* The “irrelevant for QS” listed in the original Figure 3B were the false positives without QS function. Note that users can find the details for the “irrelevant for QS” in the Supplementary Table 9.

Major comment 6 (Data provision & clarity): Figure 1 needs numbers of sequences for each of the data sets.

Response:

OK. Thanks, Done. Numbers of sequences for Dataset I~X have been added in the revised Figure 1.

Major comment 7 (Data provision & clarity): Please provide the final QSHGM as a csv/machine readable file (the web interface is nice for browsing, but without an API it would be good to share the final data in a way that facilitates future work).

Response:

OK. Thanks, Done. We named the data of the QSHGM as Dataset X, and the details have been listed in Supplementary Table 10.

Major comment 8: Why did you only use receptors for Gram positive, but synthases and receptors for gram negatives?

Response:

Good question! Firstly, we want to stress that BLASTP-based QS entries expanding and reported entries collection were combined together in this study (Figure 1) to study

the QS-based communication among the human gut microbes.

With respect to the BLASTP-based QS entries expanding, just as you pointed out, we only use receptors for Gram-positive strains, but synthases and receptors for Gram-negative strains. The reasons are listed as follows:

- (1) The signal peptide sequences of Gram-positive strains are relatively short (about 5~30 amino acids), which can easily lead to increased false positives from the BLASTP process. To improve the accuracy of the local BLASTP, we have removed the autoinducer peptides (AIPs) from Gram-positive strains.
- (2) Short peptides and proteins are not generally placed together for sequence BLASTP and functional analysis because proteins generally have a fixed structure while short peptides do not. Therefore, we specifically removed entries with length less than 100 in Dataset I to make the samples uniform and ensure the reliability of the BLASTP.

With regard to the reported entries collection, we have collected the QS&TCS entries as many as possible for the reported QS-related entries (Dataset III, 21,383 entries). Diverse autoinducer peptides of Gram-positive strains were included in the Dataset III (see more details in Supplementary Table 2), such as competence stimulating peptide AgrD and cyclic lactone autoinducer peptide.

To sum up, the synthases of the Gram-positive strains were considered in this work, but only through the reported entries collection rather than the BLASTP-based expanding to improve reliability of the latter.

Major comment 9: Machine learning: did you truncate / normalize amino acid sequences (features) to a consistent length to predict labels (negative/positive)? The text needs some improvements to clarify what was done. What exactly were features? Why did you choose multiple classifiers, all of which seem to perform similarly? How did you perform ensemble predictions (majority across the classifiers)? Can you comment on the potential that there different numbers of positive samples from Gram positive vs negative QS systems, i.e. can that confound downstream analyses?

Response:

Firstly, we would like to offer more details regarding our selection of features. The amino acid composition (AAC) calculates the frequency of each amino acid type in a protein or peptide sequence. We calculated the frequency of each amino acid type in each entry sequence as the protein features. The frequencies of all 20 natural amino

acids are the percent of the number of amino acid type divided by the length of a protein sequence [5], which is listed as follows:

$$f(t) = N(t)/N, t \in \{A, C, D, \dots, Y\} \quad (1)$$

where $N(t)$ is the number of amino acid type t , while N is the length of a protein or peptide sequence.

Note that AAC features [5] we extracted are based on the frequencies of all 20 natural amino acids, there is no need to supplement the length of protein. Therefore, we didn't normalize amino acid sequences to a consistent length for the negative or positive samples.

Secondly, we would like to explain our reasons to choose to create and utilize multiple classifiers for entries classification:

- (1) We want to mine potential QS entries for human gut microbes as inclusively as possible. Different classifiers can help us obtain different positives to mine more potential QS entries. We have manually checked their annotations and divided the union of positives into annotated positives (AP) and uncharacterized positives (UP) (Figure 3B), which were analyzed further for their specific overlaps (Figure R4, Figure S2) (see more details in Supplementary Table 12). As illustrated in the red box of Figure R4, there are 882 and 34 entries shared only by one classifier for annotated and uncharacterized positives, respectively, which indicates the need for the union of the four positives to cover more potential QS entries. Note that the union of positives from SVM and KNN classifiers are predominant, the positives from RF (65 entries for AP, 7 entries for UP) or DNN (134 entries for AP, 8 entries for UP) classifier can supplement entries to some extent. When the positives of three classifiers are combined together, such as SVM/KNN/RF and SVM/KNN/DNN, the positives from the fourth classifier will contribute even less. This indicates that the positives from the above classifiers can cover most of the entries, which is why we did not use any further classifiers.
- (2) We use these four representative machine learning models (SVM, KNN, RF, and DNN) as basic classifiers to construct the ensemble model for potential QS entries prediction. The data distribution assumption of each ideal model is different, and we used four classifiers to build an integration model that can describe different perspectives of real data. The integrated model can select more abundant and diverse proteins, overcome the over-fitting phenomenon of single model, and make

the prediction results more robustness. Therefore, the union of the four classifiers can minimize the classification bias from the scale difference of syntheses and receptors to some extent.

Figure R4. The overlaps for the AP and UP from the four ML-based classifiers.

The AAC features we calculated from the positive and negative samples were used for the training of the ensemble classifiers based on four different machine learning algorithms (SVM, RF, KNN, and DNN). “model.py” was created for training samples with SVM, KNN and RF (random forest), “nn.py” was the script used for training samples with Neural Network. Performances of the four ML-based classifiers were measured based on the accuracy, precision, recall, and F1 score. All the four classifiers were then applied to predict the entries in dataset VII are QS entries or not with the output being 1 (yes) or 0 (no), respectively. We have also added or modified some descriptions for the conduction of the four classifiers in the revised manuscript.

Finally, the numbers of positive samples from Gram-positive and Gram-negative QS systems were indeed different, but only modestly and not in the order of magnitude, which means the influence for the downstream analyses was limited. Furthermore, we did not specifically distinguish the specific entry sequence differences. We mixed them together in our proposed workflow which includes entries collecting, expanding, and mining processes. Analysis of the structure and function of a protein depends on its annotation, sequence similarity, and domains (see more details in Supplementary Table 11), regardless of whether its source is Gram-negative or Gram-positive microbes.

Major comment 10: Please justify and explain better what negative samples are. Are they aminoacids that do not have a relationship with QS, sampled from the proteomes of the named organisms? Why was this approach chosen to build a training set (as opposed to perhaps selecting from other secretion- but not QS related sequences)? I do not think anything is wrong here, necessarily, but a better explanation would help.

Response:

Thanks for your suggestion. The negative samples are the proteins that do not have a relationship with QS. We constructed negative samples by removing QS-related components from well-studied Gram-negative bacteria (*Aliivibrio fischeri*, *Escherichia coli*, *Pseudomonas aeruginosa*, *Salmonella typhimurium*, and *Vibrio parahaemolyticus*) and Gram-positive bacteria (*Bacillus subtilis*, *Staphylococcus aureus*, and *Lactococcus lactis*). More specifically, we have removed the proteins that directly and indirectly associated with QS, such as quorum sensing, luxR, two-component, homoserine-lactone synthase, histidine kinase, biofilm, autoinducer, bacteriocin, competence, virulence, signal, sensor, response, regulator, membrane, binding, transcriptional activator etc.

In order to improve the training of classifier as much as possible, the number of negative samples should be comparable to that of positive samples, and the randomness and diversity should be maintained. It is not easy to randomly obtain more than 20,000 sequences with high diversity without QS function. Therefore, we prepared negative samples based on the large dataset dataset II (818 proteomes), with the hope that (1) protein diversity of negative samples could be well guaranteed based on some whole proteomes excluding QS-related entries and (2) the accuracy of samples could be well controlled.

Major comment 11: Figure 2 (panel F in particular) needs much more annotation and explanation in the caption.

Response:

Sorry for the ambiguity. We have modified Figure 2 captions in the revised manuscript.

Major comment 12: Explain better what the communication networks are showing. E.g. Figure 5A) Does this mean both *E. coli* and *Bacteroides pectinophilus* could be ‘listening to’ and ‘speaking’ the AI-2 language? Figure 5D shows arrows and lines, and a thick white arrow, what do they mean? Please improve the clarity of the figure and annotate all that is shown in the figure or the caption.

Response:

Sorry for the ambiguity. Microbes communicate via various languages, and it is possible to predict various microbial communication and even construct a cell-cell communication network among different gut microbes based on diverse QS languages, which we termed as “QS communication network”.

Figure 5A (the revised Figure 4A) means that both *E. coli* and *Bacteroides pectinophilus* can speak the AI-2 language.

Note that the line represents that both of the strains can speak the language; black arrow means that one strain can speak the language to the other one; the thick white arrow shows that all the three strains can communicate with AHL language.

We have also supplemented the caption for the original Figure 5 (the revised Figure 4) to improve its clarity.

Major comment 13: Could you provide a computer readable table for the calculated communication networks?

Response:

OK, Thanks, Done. The details for calculated communication networks have been listed in Supplementary Table 13. We have also added the “QSCN.net” file, which can be viewed in the Pajek software.

Minor comments:

Minor comment 1: L53: re: AHL signal incorporation into plasma membranes — the reference is talking about lung epithelia. This seems not a relevant point unless also true in the gut.

Response:

Sorry for the mistake. We have replaced it with another AHL-relevant cases, which are linked to the modulation of the gut immune system.

“For example, N-(3-oxodecanoyl)-L-homoserine lactone, a common AHL-type signal, plays an important role in the modulation of the gut immune system by inducing neutrophils apoptosis [6] and attenuating innate immune responses via disruption of NF-κB signalling [7], thus providing a better colonization for *Pseudomonas aeruginosa* in the host.”

Minor comment 2: L388: Doubtful that evolution works for the good of the population.

Response:

For many years, bacterial cells were considered primarily as selfish individuals. Recently, more and more studies showed that microbes could coordinate collective behaviour in response to environmental challenges using sophisticated intercellular communication networks [8]. These cell-cell communications are mediated by quorum sensing. For example, Atkinson et al. [9] presented that bacteria are not limited to communication within their own species but are capable of listening in and broadcasting to unrelated species to intercept messages and coerce cohabitants into behavioural modifications, either for the good of the population or for the benefit of one species over another. Furthermore, Collins et al.[10] found that the indole-based communication will promote a population-based resistance mechanism constituting a form of kin selection whereby a small number of resistant mutants can, at some cost to themselves, provide protection to other, more vulnerable, cells, enhancing the survival capacity of the overall population in stressful environments. To track the emergence and fixation of each mutation in their evolution experiment, they genotyped, using mass spectrometry, their daily *E. coli* populations to estimate the allelic frequency of each single nucleotide polymorphisms. Finally, they established a population-based antibiotic-resistance mechanism based on indole as a cell-signaling molecule.

We have also added this indole-based citation to the revised manuscript to support this statement.

Minor comment 3: L361 and 439: I was confused by the use of the word “familial” and genus in the same sentence

Response:

Sorry for the confusion. We have replaced the “familial” to be “conserved”. Hope this will be better.

Response to Reviewer #2 (Expertise: complex networks, microbiome, Machine Learning, metabolic networks):

In the present work, the authors built a workflow for QS entries collecting, expanding and mining. All these QS entries are proteins. Then four ML-based models were trained to check if a protein is a QS entry. There are 21,410 positive samples were collected manually and 7,157 QS entries were predicted by ML-based models, and the total 28,567 QS entries contain 1,882 QS synthases and 26,685 receptors. Based on QS entries and QS-based microbial interactions, the authors developed the QSHGM database. Finally, a network was used to visualize the relationships among microbes based on the QSHGM database.

My major concern is how this QS-based communication network help improves our understanding of the dynamics and organizational principles of microbial ecosystems, which originally is considered largely based on metabolic interactions. The authors just briefly mentioned this in the discussion part. But a quantitative model/analysis (as an example) is needed, which is quite important.

Response:

Thanks. QS-based communication network (QSCN) is helpful for investigating and understanding the communication principles of the more sophisticated microbial communities, including cells from the same or different kingdoms [11]. As stated in our previous review (DOI: 10.1016/j.tim.2021.04.006) [8], QS can be vertically and horizontally applied to natural microbial systems and synthetic microbial consortia, respectively. Vertically, QS plays an important role in maintaining the symbiotic relationship among phages, microbes, and hosts. Horizontally, dynamic manipulations of synthetic microbiota have offered encouraging examples for further exploration of the role of QS-based interactions in the construction of diverse consortia, possibly with greater sizes and richer interactions.

Unlike metabolic interactions, most of the QS-based interactions can only improve our understanding of ecological communication principles qualitatively for now. In this work, we mainly predict the existence of various microbial communication, but we cannot give the specific intensity for them. There are mainly the following difficulties:

- (1) The reliable construction of directed QS networks including QS languages sender and receiver still faces many challenges, such as the huge network scale, multi-layer control structures, complex QS crosstalk, intricate social cheating, diverse environmental factors, different spatial distributions, and insufficient QS

entries for many uncultured microbes.

(2) The vast majority of QS-based interaction strengths are unknown, and the quantitative interactions of several QS systems were heterologously expressed in specific strains, such as *Escherichia coli* and *Salmonella typhimurium* (Table R4). Certainly, it is possible to quantitatively predict the stable synthetic microbial consortia based on these data. However, quantitative prediction of QS-based interactions in the natural microbial systems is currently difficult to achieve.

Table R4. Previous studies with quantitative QS-based interactions.

QS systems	Strains	Functions	Refs
lux	Escherichia coli	Programmed population control of an Escherichia coli population by QS-regulated killing.	[12]
lux, las	E. coli	Construct a E. coli predator–prey ecosystem.	[13]
lux, las	E. coli	Analyze the spatial and temporal dynamics of a QS-regulated synthetic, chemical-mediated ecosystem.	[14]
lux	E. coli	Construct a synchronized QS-regulated genetic clocks.	[15]
cin, rhl	E. coli	Investigate emergent genetic oscillations in a synthetic microbial consortium.	[16]
lux, las, rpa, tra	E. coli	Investigate QS communication modules for microbial consortia.	[17]
esa	E. coli	Construct a QS-based metabolic toggle switch (QS-MTS) in engineered bacteria for myo-inositol production.	[18]
lux, rpa	Salmonella typhimurium	Construct a stabilized microbial ecosystem by QS-based synchronized lysis circuits (QS-SLC).	[19]
rhl, lux, tra, las, cin, rpa	E. coli	Engineer the coordinated system behavior in synthetic microbial consortia.	[20]
lux, inducible rpa	E. coli	Construct inducible QS-regulated synthetic microbial communities.	[21]
lux, las	E. coli	Combinational quorum sensing devices for dynamic control in cross-feeding cocultivation	[22]

(3) The quantitative prediction for microbial consortia is integral to the growth dynamics of microbial communities, so it is incomplete to talk about regulation network when the growth and metabolism of microbial communities are excluded. At the same time, it is also incomplete to discuss only metabolism-based interactions while ignoring QS-based interactions for the study of microbial interactions. We will gain a more comprehensive understanding of the dynamics and organizational principles of microbial ecosystems by combining metabolic interactions and QS-based regulations. For example, Karkaria et al. [23] proposed a

methodology for designing robust synthetic communities that include competition for nutrients, and use QS to control amensal bacteriocin interactions in a chemostat environment. In our another work (DOI: 10.1016/j.ymben.2021.07.002) [22], we found that the combination of different QS devices across multiple members offers a new tool with the potential to effectively coordinate synthetic microbial consortia for achieving high product titer in cross-feeding cocultivation.

To sum up, the main outcome of this work is to reveal that microbial communication interactions exist in large quantities in diverse microbial communities by presenting the QSCN qualitatively, which is constructed based on the data collection in the QSHGM database. At the same time, we also call for more researchers to integrate metabolism-based interactions with communication-based regulations together to make the study of microbial interactions more complete. Certainly, the establishment of a more comprehensive network to achieve quantitative prediction of microbial interactions is also the priority of our next work. We have added corresponding context for the revised manuscript to stress the above points.

Major comment 1 (1.1 About the workflow): The schematic diagram of the workflow with too many different datasets, styles, colours, and unclear description is very hard to follow. The input (data source), output and filtering rules should be defined clearer for each step. A table would be quite helpful.

Response:

Thanks for the suggestion. We have revised the original Figure 1. Note that positive dataset is coloured in red; negative, green; both of them, grey. Your suggested table has been added in the Supplementary material Table S2 (Table R3 in the response to the Major comment 3 from Reviewer #1).

Major comment 2 (2.1 ML-based models): The authors used 4 different models: random forest (RF), KNN, SVM, and DNN. The authors claimed that “the RF classifier achieves the best performance” based on accuracy and F1 score (line 204). Why did the authors use all four models instead of the best model?

Response:

There are two reasons for us to choose to create and utilize multiple classifiers for entries classification, which have been detailed in the response to the Major comment 9 from Reviewer #1.

Major comment 3 (2.2 ML-based models): The authors mentioned “four ensemble classifiers” (line 126). How did the authors combine 4 different algorithms together by ensemble learning?

Response:

Sorry for the ambiguity. As illustrated in the revised Figure 1, the four classifiers were utilized to classified the dataset VII (9,253 entries), respectively. We take the union of the positives obtained by the four classifiers to obtain more extended QS entries. The union of the positives was then divided into uncharacterized (Dataset VIII) and annotated positives. The former was re-annotated, mined, and sorted out manually with the help of UniProt, NCBI, and Phyre². The latter was combined with the re-annotated positives from Dataset VIII to decide whether the entries have QS function or not by function analysis, which was based on the specific annotations, sequence similarity, and domains (more details in Supplementary Table 11).

Major comment 4 (2.3 ML-based models): There are 21,410 positive samples (line 248), but the number of total samples (proteins) in 818 gut microbes and the number of negative samples is unclear. Is a sample either positive sample or negative sample?

Response:

This seems to be a concern also shared by Reviewer #1. There are 22,780 entries we collected as the negative samples. More details for negative samples have been given in the response to the Major comment 10 from Reviewer #1.

Major comment 5 (2.4 ML-based models): The authors mentioned that 21,410 positive samples were collected manually. The search was based on four commonly used QS annotations, i.e., “quorum sensing”, “LuxR”, “two-component”, and “tryptophanase” (line 131). Do all “two-component regulatory systems” work as quorum sensing? Are these four keywords specific enough to narrow the searching scope and reduce the false-positive QS entry?

Response:

Sorry for the ambiguity. Generally, various QS systems can be roughly divided into three types: (i) acylated homoserine lactones (AHLs) and other autoinducers received by LuxR-type receptors utilized by Gram-negative bacteria; (ii) auto-inducing peptides (AIPs) and other autoinducers sensed by two-component systems (TCSs) utilized by Gram-positive bacteria; and (iii) autoinducer 2 (AI-2) and indole for interspecies communication of microbial communities [3]. “quorum sensing” can cover some

entries with obvious QS-based function; “LuxR” can cover a very common type of QS receptor; “two-component” can cover TCSs, which play an important role in microbial communications; “tryptophanase” represents for indole-based communication. The dataset created based on these four keywords is big enough to cover the data from SigMol, Quorumpeps, and other reported QS entries.

TCS entries play an important role in microbial communications, which have a certain overlap on QS [2], but it is difficult to separate them clearly for now. In this study, due to the intricate overlaps on QS and TCS entries, we collected the TCS entries as one of the important parts for the reported QS&TCS entries of 818 human gut microbes. The specific reasons have been listed in the response to the Major comment 2 from Reviewer #1.

Major comment 6 (2.5 ML-based models): The authors mentioned that “All the four classifiers were applied to predict that whether the input amino acid sequences are QS entries or not with the output being 1 or 0, respectively” (line 488). However, the input for each algorithm, which is how to represent an amino acid sequence with variable length, was not mentioned in the draft.

Response:

This is again a point shared by Reviewer #1. The the Amino Acid Composition (AAC) calculates the frequency of each amino acid type in a protein or peptide sequence. We calculated AAC in each entry sequence as the protein features (more details for this have been listed in the response to the Major comment 9 from Reviewer #1).

Major comment 7 (2.6 ML-based models): There are two different QS entry types: QS synthase and receptor, and the total 28,567 QS entries contain 1,882 QS synthases and 26,685 receptors. Receptors must also dominate the training set. Is there any bias when we predict QS synthases since the sequences of QS synthases may be different from receptors’ dramatically?

Response:

Thanks. We admit that due to the differences in the scale of the corresponding data used for training, there will be potential deviations in the accuracy of the classifier in identifying the corresponding entries. The union of the four classifiers we constructed can minimize the classification bias from the scale difference of synthases and receptors to some extent (More details have been listed in the response to the Major comment 9 from Reviewer #1).

Furthermore, it is expected for the number of receptors to be more than synthases, which is not a problem of our data. There are many cases in nature where there are only receptors but no synthases in one specific organism. For example, although most *E. coli* have acylated homoserine lactones (AHL) receptor (SdiA), they lack AHL synthase. Therefore, the uneven data of QS synthases and receptors coincides with their natural distribution.

Major comment 8 (3.1 QS-based microbial interactions): The authors claimed that “according to the collected data in the QSHGM database, we can predict various potential pairwise QS-based microbial interactions” (line 289) and “predict more sophisticated interaction networks” (line 298). However, the details of this prediction method cannot be found in this draft. Did the relation between two microbes include in the database? If so, the data source should be pointed out.

Response:

Thanks for the suggestion. We mean that the new QS-based communication interaction can be searched in our QSHGM database. To make it easier, we have illustrated these interactions in our QSCN. The details for calculated communication networks have been listed in Supplementary Table 13. We have also added the “QSCN.net” file, which can be viewed in the Pajek software.

Major comment 9 (3.2 QS-based microbial interactions): According to Fig. 7B, one microbe (such as *E. coli*) can produce compounds (such as CAI-1) by QS synthases, and another microbe (such as *Bacteroides vulgatus*) can receive the same compound by its receptor, so they can communicate with each other. Since each microbe can produce so many compounds and each microbe also can receive so many different compounds, Fig. 7A gives a complete graph that links every node to every other node for the 40 core microbes. We cannot figure out that what information this graph can provide.

Response:

Sorry for the ambiguity. The QSCN we constructed (Figure 5A) can be projected to a one-mode network (Figure S4) that visualizes microbial communication-based interactions directly. This one-mode network is characterized by a large number of highly connected nodes. To obtain the core microbes in our proposed QSCN, we shrunk the network to get key nodes with large degree and high betweenness centrality. Therefore, such a dense network (original Figure 7A) was provided to indicate the 40 core gut microbes with excellent microbial communications. As such, what is

visualized here is essentially a sub-network with a particularly high “density”, not representative of the entire network of the whole gut environment. We would also like to point out that, although each microbe can produce so many QS languages in the 40 core gut microbes, the specific intensity for each language cannot be provided in this work; the intention is that we first determine the existence of the QS-based communications (as done in this work), and then to investigate its corresponding intensity (future work), eventually bringing a comprehensive understanding of the communication-based microbial interactions.

To avoid misunderstanding, we have moved it from the revised manuscript into the supplementary material as Figure S5. We have also added some corresponding context in the revised manuscript.

Major comment 10 (3.3 QS-based microbial interactions): Same concern as 2.4 and described in 3.2, there are so many links between each different species. The false-positive relations between each different species should be tackled carefully.

Response:

Thanks. We admit that there will be inevitably some false positive entries in our QSHGM database and the proposed QSCN, even if there is no problem at the level of individual nodes, the relations we have predicted were not necessarily always true in reality. On the one hand, we want to stress that manual curation, BLASTP-based expanding, and ML-based classifications were combined together in this work to minimize false positives as possible. On the other hand, QS links we predicted based on the database would be “possibilities”, not reality and still require experimental verification. However, we offered a tool to allow users with various applications in mind to see the “possibilities” in the first place, which allows them to subsequently focus their experimental verification.

Furthermore, we have added some corresponding statements for this concern in the revised discussion section of the new manuscript.

References

1. Meng, C. et al. (2020) CWLy-SVM: A support vector machine-based tool for identifying cell wall lytic enzymes. *Comput Biol Chem* 87, 107304.
2. Wang, S. et al. (2020) Quorum Sensing Communication: Molecularly Connecting Cells, Their Neighbors, and Even Devices. *Annual Review of Chemical and Biomolecular Engineering* 11 (1), 447-468.
3. Wu, S. et al. (2020) Quorum sensing for population-level control of bacteria and potential therapeutic applications. *Cell Mol Life Sci* 77 (7), 1319-1343.
4. Bernard, C. et al. (2020) Rich Repertoire of Quorum Sensing Protein Coding Sequences in CPR and DPANN Associated with Interspecies and Interkingdom Communication. *mSystems* 5 (5).
5. Chen, Z. et al. (2018) iFeature: a Python package and web server for features extraction and selection from protein and peptide sequences. *Bioinformatics* 34 (14), 2499-2502.
6. Tateda, K. et al. (2003) The *Pseudomonas aeruginosa* autoinducer N-3-oxododecanoyl homoserine lactone accelerates apoptosis in macrophages and neutrophils. *Infect Immun* 71 (10), 5785-93.
7. Kravchenko VV et al. (2008) Modulation of gene expression via disruption of NF- κ B signaling by a bacterial small molecule. *Science*.
8. Wu, S. et al. (2021) Vertical and horizontal quorum-sensing-based multicellular communications. *Trends Microbiol*.
9. Atkinson, S. and Williams, P. (2009) Quorum sensing and social networking in the microbial world. *J. R. Soc. Interface* 6 (40), 959-978.
10. Lee, H.H. et al. (2010) Bacterial charity work leads to population-wide resistance. *Nature* 467 (7311), 82-85.
11. Whiteley, M. et al. (2017) Progress in and promise of bacterial quorum sensing research. *Nature* 551 (7680), 313-320.
12. You, L. et al. (2004) Programmed population control by cell-cell communication and regulated killing. *Nature* 428 (6985), 868-871.
13. Balagadde, F.K. et al. (2008) A synthetic *Escherichia coli* predator-prey ecosystem. *Mol Syst Biol* 4, 187-195.
14. Song, H. et al. (2009) Spatiotemporal modulation of biodiversity in a synthetic chemical-mediated ecosystem. *Nat Chem Biol* 5 (12), 929-935.
15. Danino, T. et al. (2010) A synchronized quorum of genetic clocks. *Nature* 463 (7279), 326-30.
16. Chen, Y. et al. (2015) SYNTHETIC BIOLOGY. Emergent genetic oscillations in a synthetic microbial consortium. *Science* 349 (6251), 986-9.
17. Scott, S.R. and Hasty, J. (2016) Quorum Sensing Communication Modules for Microbial Consortia. *Acs Synth Biol* 5 (9), 969-977.
18. Gupta, A. et al. (2017) Dynamic regulation of metabolic flux in engineered bacteria using a pathway-independent quorum-sensing circuit. *Nat Biotechnol* 35 (3), 273-279.
19. Scott, S.R. et al. (2017) A stabilized microbial ecosystem of self-limiting bacteria using synthetic quorum-regulated lysis. *Nat Microbiol* 2, 17083.
20. Kylilis, N. et al. (2018) Tools for engineering coordinated system behaviour in synthetic microbial consortia. *Nat Commun* 9 (1), 2677.
21. Miano, A. et al. (2020) Inducible cell-to-cell signaling for tunable dynamics in microbial communities. *Nat Commun* 11 (1), 1193.
22. Wu, S. et al. (2021) Combinational quorum sensing devices for dynamic control in cross-feeding cocultivation. *Metab Eng* 67, 186-197.
23. Karkaria, B.D. et al. (2021) Automated design of synthetic microbial communities. *Nat Commun* 12 (1), 672.

Reviewers' Comments:

Reviewer #1:

Remarks to the Author:

The authors have addressed most of my comments and improved the manuscript. I now recommend publication with some minor improvements.

- A comment on my previous "Major point 8: Why did you only use receptors for Gram positive, but synthases and receptors for gram negatives?" I understand the explanation provided. However, false positives due to short length of the peptides could be accounted for differently, instead of removing them entirely. For starters, a lower e-value threshold could have been chosen to only allow for exact matches; secondly, exact matches that are otherwise unlikely genes for small peptide production (e.g. occurring within an ORF) could be removed. I do not expect the authors to change their methodology, but explain their choices (for the e-value threshold applied, and to remove the peptides entirely rather than curate them) in the methods. However, these points lead to an actionable request: please address in the discussion of the communication networks that a class of signals were excluded and thus did not inform the networks inferred.

- In the discussion, mention that the machine learning methods were run on AA frequencies. Explain and discuss that potentially important information on the AA sequence is lost and not considered by the machine learning tools. What that might that imply? This seems particularly important for short sequences.

- Explain the output of "Positive and negative samples construction." at the end of the method section: Presumably, an array of 22,780 AAs FREQUENCIES found in 8 organisms calculated from the AA sequences of proteins after removing QS related proteins.

- Regarding reply to minor comment 2: the sentence on line 383, "...'broadcasting' to unrelated species for the good of the population" is implausible as an evolutionary statement. It is unsubstantiated by the presented work because no experiments were performed. And it does not follow from the references, which study a population of a single species.

Reviewer #2:

Remarks to the Author:

I appreciate the improvement and major revision of the manuscript by the authors, especially on the role and limitation of QSCN in understanding the dynamics and organizational principles of microbial ecosystems. Most of the questions I asked are much clearer now. I anticipate that the authors will be further encouraged to explain or address the following issues to improve the manuscript and its understanding to a broad readership and quantify the claims.

1. QSCN – QS language

Since QSCN is one of the most important contributions in this article, it's better to clarify the hypothesis and related definitions clearly in this manuscript instead of letting the reader read the previously published work.

The authors defined 9 types of QS language (Line 46, essentially they are small molecules or peptides). It looks like "QS signals" has the same meaning as "QS language". Two of them involved in the communication of interspecies (AI-2 and indole) and the others involved in the communication of intraspecies (L46 – L50). The amount of QS language can affect the QSCN dramatically. Let's say there is only one QS language "QSL", all of the related QS entries can be connected by QSL and an extremely simple network will be built. If there are 100 different QS languages, however, this network will become much more complicated. I would like to ask:

- a) How did the authors define 9 types of QS language?
- b) Is there any other QS languages?

2. QSCN – The hypothesis in each talk

As far as I understand, a single communication (talk) should have both QS signals producer

(maybe related enzyme) and QS signals receiver (maybe related receptor). Same example in Fig. 6 (previous Fig. 7B): one microbe (such as *E. coli*) can produce compounds (such as CAI-1) by QS synthases, and another microbe (such as *Bacteroides vulgatus*) can receive the same compound by its receptor. So *E. coli* can speak to *B. vulgatus* and each talk should be directional naturally. I would like to ask why QSCN is not a directed graph.

Furthermore, how to differentiate a bacteria as a producer or receiver is unclear. When I search "AI-2" in QSHGM website, I can find 567 QS entries. The first entry is W1Q6C6 and the annotation in UniProt of this entry is: "Involved in the synthesis of autoinducer 2 (AI-2) which is secreted by bacteria and is used to communicate both the cell density and the metabolic potential of the environment ...". In my understanding, the strains (or species) that can express protein W1Q6C6 should be a producer (whether they are receivers is unclear). However, the authors connected all these producers together by AI-2 in Fig. 5A and explained further in Fig. S7c: "One can find that ten members of the community can communicate based on the AI-2 language (Figure S7c), thus leading to a potentially dense QSCN." (L100 – L102 in supplementary material). I also noticed the response of "major comment 12" from the first reviewer: "Note that the line represents that both of the strains can speak the language; black arrow means that one strain can speak the language to the other one". I would like to ask how they can talk with each other if there are only producers and devoid of receivers. It seems like that the authors had a strong hypothesis when the network was built: all producers (speakers) should be receivers to the same QS language as well. Is this hypothesis reasonable? Is there any other hypothesis didn't describe explicitly?

One more question about QSCN. There are 2 different types of QS languages, one for the communication of interspecies and another for the communication of intraspecies. They may have different abilities of connecting gut microbes. But I cannot see any different connection patterns in Fig. 5A.

3. ML – the evaluation of model performance

According to my experience, the evaluation of model performance should be depended on the test set or cross-validation first. Since the authors used 5-fold cross-validation on the whole dataset (21,383 positive samples in Dataset III and 22,780 negative samples in Dataset IV), 5 different accuracies (same to precision, recall and F1 score) should be got and finally the average accuracy (same to precision, recall and F1 score) should be reported. Once the model was well-trained, it can be applied to other datasets (model prediction), such as uncharacterized positives (Dataset VII).

The authors explained how the rate of false positives was addressed on pages 1-2 of "to editor" part. However, the prediction result of Dataset VII was discussed simultaneously. It makes me confused about how the model was evaluated. Was the reported accuracy calculated based on the result of 5-fold cross-validation or based on the re-annotated result (the union of the positives in the prediction result of Dataset VII)?

I noticed that the authors stated that "Classifiers were trained and validated based on the positive and negative samples, and then tested on the dataset V (Fig. 2)." (L563-L564). Another thing I would like to point out is that Dataset V is not a valid test set, since not all of the QS entries were determined as positive or negative samples. Only if the re-annotation steps did before model prediction (all of the QS entries in Dataset V were determined as positive samples or not), Dataset V can be used as a valid test set.

4. ML – the key hyper-parameters

How did the authors determine the key hyper-parameters, such as the number of layers and learning rate in the DNN model and SVM? I noticed that only the hyper-parameter in the KNN model was mentioned (L573 – L604).

Dear reviewers,

Machine Learning Aided Construction of the Quorum Sensing Communication Network for Human Gut Microbiota

Shengbo Wu, Jie Feng, Chunjiang Liu, Hao Wu, Zekai Qiu, Jianjun Ge, Shuyang Sun, Xia Hong, Yukun Li, Xiaona Wang, Aidong Yang, Fei Guo, and Jianjun Qiao.

We are very grateful for the valuable comments from reviewers, which have now all been carefully considered in the revision of our manuscript. In the revised manuscript, all the changes are marked in red.

The following is the point-to-point response to individual comment marked in blue.

Response to Reviewer #1:

Reviewer #1 (Remarks to the Author):

The authors have addressed most of my comments and improved the manuscript. I now recommend publication with some minor improvements.

Response:

Thanks for your interest in this work. We have further revised the manuscript carefully.

Minor comment 1:

- A comment on my previous “Major point 8: Why did you only use receptors for Gram positive, but synthases and receptors for gram negatives?” I understand the explanation provided. However, false positives due to short length of the peptides could be accounted for differently, instead of removing them entirely. For starters, a lower e-value threshold could have been chosen to only allow for exact matches; secondly, exact matches that are otherwise unlikely genes for small peptide production (e.g. occurring within an ORF) could be removed. I do not expect the authors to change their methodology, but explain their choices (for the e-value threshold applied, and to remove the peptides entirely rather than curate them) in the methods. However, these points lead to an actionable request: please address in the discussion of the communication networks that a class of signals were excluded and thus did not inform the networks inferred.

Response:

Thanks for the good suggestion. We have revised the corresponding content in both the “Discussion” part and the “Data acquisition” part in the “Methods” section.

Minor comment 2:

In the discussion, mention that the machine learning methods were run on AA frequencies. Explain and discuss that potentially important information on the AA sequence is lost and not considered by the machine learning tools. What that might that imply? This seems particularly important for short sequences.

Response:

Thanks for the good suggestion. We have added some sentences in the revised “Discussion” part marked in red. Hope these will be better.

The four ML-based classifiers were trained on amino acids frequencies, which did not include the physicochemical properties (such as hydrophobicity, charge and molecular size) ¹ and the information on amino acids combinations with fixed length ². The above features are particularly important for the accurate prediction of the short sequences such as the AIPs (about 5~30 amino acids).

Note that short peptides (such as AIPs) and proteins are not generally placed together for sequence BLASTP and functional analysis, because proteins generally have a fixed structure while short peptides do not. AIPs sequences can also easily lead to increased false positives from the BLASTP process. Therefore, to increase the reliability of the expansion, we have removed the signal peptides in the BLASTP-related datasets (I and VII), thus leading to the sparse edges for the “AIPs” node in our QSCN (Fig. 5A). This calls for a more accurate method to cover more aforementioned amino acids features for short sequences to mine the potential signal peptides in the future to make the QSCN more complete.

Combined with excellent suggestions from comments 1 and 2, we have stressed the above discussion about the future improvements for QSCN in the expansion and prediction of AIPs, which will be the focus of our next work in the near future.

Minor comment 3:

Explain the output of “Positive and negative samples construction.” at the end of the method section: Presumably, an array of 22,780 AAs FREQUENCIES found in 8 organisms calculated from the AA sequences of proteins after removing QS related proteins.

Response:

Thanks for the advice. We have added further clarifications at the “Positive and negative samples construction” section.

Minor comment 4:

Regarding reply to minor comment 2: the sentence on line 383, “...’broadcasting‘ to unrelated species for the good of the population” is implausible as an evolutionary statement. It is unsubstantiated by the presented work because no experiments were performed. And it does not follow from the references, which study a population of a single species.

Response:

Thanks for the suggestion. We have removed this inappropriate description in the

revised manuscript.

Response to Reviewer #2 (Remarks to the Author):

I appreciate the improvement and major revision of the manuscript by the authors, especially on the role and limitation of QSCN in understanding the dynamics and organizational principles of microbial ecosystems. Most of the questions I asked are much clearer now. I anticipate that the authors will be further encouraged to explain or address the following issues to improve the manuscript and its understanding to a broad readership and quantify the claims.

Response:

Thanks for good suggestions on our work. We have revised the manuscript carefully.

Comment 1 (QSCN-QS language): Since QSCN is one of the most important contributions in this article, it's better to clarify the hypothesis and related definitions clearly in this manuscript instead of letting the reader read the previously published work.

Response:

Thanks. We have now considered this together with your other Comments, and revised the corresponding descriptions in the new manuscript.

Comment 2 (QSCN-QS language): The authors defined 9 types of QS language (Line 46, essentially they are small molecules or peptides). It looks like "QS signals" has the same meaning as "QS language". Two of them involved in the communication of interspecies (AI-2 and indole) and the others involved in the communication of intraspecies (L46 – L50). The amount of QS language can affect the QSCN dramatically. Let's say there is only one QS language "QSL", all of the related QS entries can be connected by QSL and an extremely simple network will be built. If there are 100 different QS languages, however, this network will become much more complicated. I would like to ask:

- a) How did the authors define 9 types of QS language?
- b) Is there any other QS languages?

Response:

Generally, microbes communicate via various QS signals, which are also termed (informally) as microbial languages by other researches³⁻⁵. Same as in these previous studies, "QS signals" has the same meaning as "QS languages" In this work.

Regarding the number of QS languages, it is to a large extent a matter reflecting the progress in QS research which has evolved from the initial work by Fuqua et al.⁶ (in 1994) on the discovery of cell-density-dependent bioluminescence. Subsequently, as summarized in the Table R1 (Table S2), various QS signals (QS languages), such as acyl-homoserine lactones (AHLs) and diffusible signaling factors (DSFs) were found in various bacterial cells⁷. In this work, based on a large amount of the reported

lectures and some existing databases (SigMol⁸ and Quorumpeps⁹), we decided to focus on the commonly reported nine types of QS languages, i.e., AHLs, DSFs, HAQs, CAI-1, AIPs, Photopyrones, Dialkylresorcinols, indole, and AI-2. Due to the lack of the sequence information for the corresponding synthases, which is needed by the approach taken in this work, some other QS languages, such as autoinducer-3 (AI-3)¹⁰, were not considered.

Table R1. Details for the selected nine types of QS languages

Type	Languages	Acronym	Synthase	Reported microbes	Refer.
Intra-species	Acyl-homoserine lactones	AHLs	Diverse I proteins (e.g., LuxI)	Most of Gram-negative bacteria (e.g., Vibrio fischeri)	6
	Diffusible signal factors	DSFs	RpfF	Xanthomonas campestris	5
	4-hydroxy-2-alkylquinoline	HAQs	PqsA	Pseudomonas aeruginosa	11
	Cholera autoinducer 1	CAI-1	CqsA	Vibrio spp.	12
	Dialkylresorcinols	DARs	DarB	Photorhabdus asymbiotica	13
	Photopyrones	-	PpyS	Photorhabdus luminescens	14
	Auto-inducing peptides	AIPs	Synthases of signal peptides (e.g., NisA)	Most of Gram-positive bacteria (e.g., Lactococcus lactis)	15
Inter-species	Indole	-	TnaA	Some microbes (e.g., Escherichia coli)	16
	Autoinducer 2	AI-2	LuxS	Most of microbes (e.g., Vibrio harveyi)	17

New QS languages are not always easy to be discovered, because they tend to autoregulation (i.e., languages positively regulate their own activity via expression of their synthase) and binding to their corresponding receptor to form complexes. Nevertheless, it is foreseeable that some novel QS languages belonging to various chemical classes will be discovered and identified in the future. Although the current work has been based on the above nine languages, we will gradually update the QSHGM and the QSCN with new discoveries.

To highlight the limitation of considering the specific nine QS languages, we have

added some notes to the section of “QS communication network construction” and “Discussion”, in the revised manuscript.

Comment 3 (QSCN – The hypothesis in each talk): As far as I understand, a single communication (talk) should have both QS signals producer (maybe related enzyme) and QS signals receiver (maybe related receptor). Same example in Fig. 6 (previous Fig. 7B): one microbe (such as *E. coli*) can produce compounds (such as CAI-1) by QS synthases, and another microbe (such as *Bacteroides vulgatus*) can receive the same compound by its receptor. So *E. coli* can speak to *B. vulgatus* and each talk should be directional naturally. I would like to ask why QSCN is not a directed graph.

Response:

The QSCN shown in Fig. 6 is actually a directed graph. The accurate QSCN for human gut microbes should also be a directed graph including QS languages producing and receiving. However, there are many challenges to construct the accurate directed large-scale QSCN. We have made some revisions in the new manuscript to stress the direction properties and challenges for the construction of the accurate QSCN.

In this study, we developed the QSHGM database to bridge the gap between QS repositories and human gut microbiota. With the help of QSHGM, the QSCN could be further constructed for human gut microbes. As stated in the third paragraph of first part of the “Discussion” section, by differentiating QS signals producing and receiving with the help of both QS synthases and receptors, there is potential to construct a directed and more precise QSCN. However, the reliable construction of the precise QSCN still faces many challenges, such as the huge network scale, multi-layer control structures, complex QS crosstalk, intricate social cheating, diverse environmental factors, different spatial distributions, and insufficient QS entries for many uncultured microbes. Therefore, we firstly constructed a preliminary and undirected QSCN based on the connections of nine QS languages (AHLs, DSFs, HAQs, CAI-1, AIPs, Dialkylresorcinols, Photopyrones, indole, and AI-2) without differentiating QS signals producing and receiving. As you can see, while pointing out the challenge of constructing a complete QSCN, we used a simplified 7-strain model from Colosimo et al¹⁸ as an example (Fig. 6) to illustrate the directed properties of the QSCN. Based on the prediction of QS languages producing and receiving, more future work will be conducted to develop a directed and complete QSCN, including as many QS languages and receptors as possible.

Comment 4 (QSCN – The hypothesis in each talk): Furthermore, how to differentiate a bacteria as a producer or receiver is unclear. When I search “AI-2” in QSHGM website, I can find 567 QS entries. The first entry is W1Q6C6 and the annotation in UniProt of this entry is: “Involved in the synthesis of autoinducer 2 (AI-2) which is

secreted by bacteria and is used to communicate both the cell density and the metabolic potential of the environment ...”. In my understanding, the strains (or species) that can express protein W1Q6C6 should be a producer (whether they are receivers is unclear). However, the authors connected all these producers together by AI-2 in Fig. 5A and explained further in Fig. S7c: “One can find that ten members of the community can communicate based on the AI-2 language (Figure S7c), thus leading to a potentially dense QSCN.” (L100 – L102 in supplementary material). I also noticed the response of “major comment 12” from the first reviewer: “Note that the line represents that both of the strains can speak the language; black arrow means that one strain can speak the language to the other one”. I would like to ask how they can talk with each other if there are only producers and devoid of receivers. It seems like that the authors had a strong hypothesis when the network was built: all producers (speakers) should be receivers to the same QS language as well. Is this hypothesis reasonable? Is there any other hypothesis didn't describe explicitly?

Response:

Sorry for the ambiguity on our hypothesis. Indeed, there were some hypotheses in the construction of the undirected and preliminary QSCN, which was based on the connections of nine types of QS languages.

QS is a cell-cell communication mechanism that consists of QS signal producing by QS synthase (such as LuxI) and signal receiving by its corresponding QS receptor (such as LuxR). It indicates that when a microbe owns a complete QS system, it can play the role of speaker and receiver to the same language at the same time. However, as we replied in the Comment 3, it is difficult to match the receptor and QS language for the gut microbiome due to the multi-layer control structures, complex QS crosstalk, and other complexities associated with the 818 human gut microbes. Although it is known that some specific microbial cheaters only have orphan LuxR-type QS receptors, such as PluR and PauR, without the corresponding AHLs synthase¹⁹, such information is still far from complete. Note that in the context of microbial social evolution, one process that stabilizes cooperation is to avoid the displacement of cooperative cells by noncooperative cheater mutants²⁰. This suggests that most of microbes contain a complete QS system compared to cheaters that only contain part of the QS modules. In the lack of more complete information, we therefore think that it is reasonable to hypothesize in the present work that all producers (speakers) should be receivers to the same QS language as well.

To make such hypotheses more explicit, we have now introduced a table of the relevant hypotheses and future improvements for this work (see Discussion and Table S4 which is shown below as Table R2). Among the five hypotheses, the first four have been explained and listed in the revised manuscript and supplementary materials. The

last one is about the hypothesis of QS signal crosstalk, that is, considering the QS crosstalk widely exists in nature ²¹, we hypothesize that species using (same or different) QS signal molecules within the same type of QS language, such as AHLs, can communicate with each other.

Table R2. Hypotheses and future improvements for the QSCN

Items	Hypotheses	Future improvement for QSCN
Microbe	Human gut microbiome consists of 818 microbes from VMH database.	Enlarge the number and range of gut microbes.
Language	There are nine types of QS languages.	Develop the QSCN for more AIPs and novel QS languages.
TCS	TCS entries possess QS functionality.	Figure out the differences and connections between QS and TCS.
Cheating	All producers are also receivers to the same QS language.	Construct a directed and more accurate QSCN differentiating QS languages producing and receiving.
QS Crosstalk	Microbes that speak the same type of languages can communicate with each other.	Quantify the intensity of QS crosstalk for the same type of QS languages to develop a weighted QSCN.

In the content of the new manuscript and supporting material, we have provided the hypotheses on cheating and crosstalk, discussed the potential future improvements for the QSCN (Table S4), and rearranged the corresponding content in a clearer layout.

Comment 5 (QSCN – The hypothesis in each talk): One more question about QSCN. There are 2 different types of QS languages, one for the communication of interspecies and another for the communication of intraspecies. They may have different abilities of connecting gut microbes. But I cannot see any different connection patterns in Fig. 5A.

Response:

Thanks for the helpful comment. We have now revised Fig. 5 to add two patterns for generally recognized intra-species (AHLs, DSFs, HAQs, CAI-1, AIPs, Dialkylresorcinols, and Photopyrones) or inter-species (indole and AI-2) QS languages. The former was marked in blue, and the latter was marked in red. As a result, there are significant genus-level overlaps between microbes on what are commonly regarded as intra-species languages, which suggests that these languages may also be involved in some interspecies communications. This indicates that the distinction between intra-species and inter-species communication languages can

become blurred, and there is no significant difference between the two patterns in the undirected and preliminary QSCN as constructed in this present work.

Comment 6 (ML – the evaluation of model performance): According to my experience, the evaluation of model performance should be depended on the test set or cross-validation first. Since the authors used 5-fold cross-validation on the whole dataset (21,383 positive samples in Dataset III and 22,780 negative samples in Dataset IV), 5 different accuracies (same to precision, recall and F1 score) should be got and finally the average accuracy (same to precision, recall and F1 score) should be reported. Once the model was well-trained, it can be applied to other datasets (model prediction), such as uncharacterized positives (Dataset VII).

The authors explained how the rate of false positives was addressed on pages 1-2 of “to editor” part. However, the prediction result of Dataset VII was discussed simultaneously. It makes me confused about how the model was evaluated. Was the reported accuracy calculated based on the result of 5-fold cross-validation or based on the re-annotated result (the union of the positives in the prediction result of Dataset VII)?

Response:

Sorry for the confusion. Indeed, as you stated above and also as illustrated in Fig. 1, we conducted 5-fold cross-validation on the whole dataset (21,383 positive samples in Dataset III and 22,780 negative samples in Dataset IV), calculated the average accuracy (same to precision, recall and F1 score), and listed them in Fig. 3A. Then, the uncharacterized dataset VII was predicted and classified by the trained ML-based classifiers.

In the previous response to the editor, what we wanted to stress is the classified results from the four well-trained ML-based classifiers and the process followed.

We have now revised the manuscript to distinguish the model training and application process to avoid misunderstanding in the section of “Expanded and new QS entries”.

Comment 7 (ML – the evaluation of model performance): I noticed that the authors stated that “Classifiers were trained and validated based on the positive and negative samples, and then tested on the dataset V (Fig. 2).” (L563-L564). Another thing I would like to point out is that Dataset V is not a valid test set, since not all of the QS entries were determined as positive or negative samples. Only if the re-annotation steps did before model prediction (all of the QS entries in Dataset V were determined as positive samples or not), Dataset V can be used as a valid test set.

Response:

Sorry for the mistake. As stated in the response to Comment 6, the four ML-based

classifiers were applied to the classification in Dataset VII. Indeed, after excluding from Dataset V the entries which were already collected as the reported QS&TCS entries in dataset III (Dataset VI, 5,320 entries), the remaining entries (Dataset VII, 9,253 entries) were then classified by the four ML-based classifiers. We have revised the manuscript to correct the mistake in the section of “ML-based classifiers” of “Method”.

Comment 8 (ML – the key hyper-parameters): How did the authors determine the key hyper-parameters, such as the number of layers and learning rate in the DNN model and SVM? I noticed that only the hyper-parameter in the KNN model was mentioned (L573 – L604).

Response:

For SVM and RF models, we used GridSearchCv²² to select and determine the optimal combination of hyper-parameters automatically to achieve best performance.

For the DNN model, as shown in the “Methods” section, we have added two hidden layers after the one-to-one layer. The first hidden layer is fully connected with one-to-one layer and the second hidden layer is fully connected with the first hidden layer. The last layer is an output layer which has two neurons. We use *softmax* as the output function of the last layer in neural networks, which turns the score produced by the neural network into values that can solve our problem. Batch normalization was applied to the one-to-one layer and each hidden layer to accelerate the training process. The SGD optimizer was used to train the DNN model and the learning rate was fixed as 0.01. Values of the other hyper-parameters of the DNN model were set to default ones without tuning.

We have added some content to the end part of each ML-based algorithm in “Methods” section to provide the above information on the generation and selection of key hyper-parameters for the four ML models.

References

1. Lee TY, Lin ZQ, Hsieh SJ, Bretana NA, Lu CT. Exploiting maximal dependence decomposition to identify conserved motifs from a group of aligned signal sequences. *Bioinformatics* **27**, 1780-1787 (2011).
2. Chen Z, *et al.* Ifeature: A python package and web server for features extraction and selection from protein and peptide sequences. *Bioinformatics* **34**, 2499-2502 (2018).
3. Brameyer S, Bode HB, Heermann R. Languages and dialects: Bacterial communication beyond homoserine lactones. *Trends Microbiol.* **23**, 521-523 (2015).
4. Shi YM, Bode HB. Microbiology: A new language for small talk. *Nat. Chem. Biol.* **13**, 453-454 (2017).
5. Deng YY, Wu JE, Tao F, Zhang LH. Listening to a new language: Dsf-based quorum sensing in gram-negative bacteria. *Chem. Rev.* **111**, 160-173 (2011).
6. Fuqua WC, Winans SC, Greenberg EP. Quorum sensing in bacteria: The luxR-luxI family of cell density-responsive transcriptional regulators. *J. Bacteriol.* **176**, 269-275 (1994).
7. Wu S, Xu C, Liu J, Liu C, Qiao J. Vertical and horizontal quorum-sensing-based multicellular communications. *Trends Microbiol.* **29**, 1130-1142 (2021).
8. Rajput A, Kaur K, Kumar M. Sigmol: Repertoire of quorum sensing signaling molecules in prokaryotes. *Nucleic Acids Res.* **44**, 634-639 (2016).
9. Wynendaele E, *et al.* Quorumpeps database: Chemical space, microbial origin and functionality of quorum sensing peptides. *Nucleic Acids Res.* **41**, D655-659 (2013).
10. Sperandio V, Torres AG, Jarvis B, Nataro JP, Kaper JB. Bacteria-host communication: The language of hormones. *Proc Natl Acad Sci U S A* **100**, 8951-8956 (2003).
11. Déziel E, *et al.* Analysis of pseudomonas aeruginosa 4-hydroxy-2-alkylquinolines (haqs) reveals a role for 4-hydroxy-2-heptylquinoline in cell-to-cell communication. *Proc Natl Acad Sci U S A* **101**, 1339-1344 (2004).
12. Higgins DA, Pomianek ME, Kraml CM, Taylor RK, Semmelhack MF, Bassler BL. The major vibrio cholerae autoinducer and its role in virulence factor production. *Nature* **450**, 883-886 (2007).
13. Brameyer S, Kresovic D, Bode HB, Heermann R. Dialkylresorcinols as bacterial signaling molecules. *Proceedings of the National Academy of Sciences* **112**, 572-577 (2015).
14. Brachmann AO, *et al.* Pyrones as bacterial signaling molecules. *Nat. Chem. Biol.* **9**, 573-578 (2013).
15. Monnet V, Juillard V, Gardan R. Peptide conversations in gram-positive bacteria. *Crit. Rev. Microbiol.* **42**, 339-351 (2014).
16. Wang D, Ding X, Rather PN. Indole can act as an extracellular signal in escherichia coli. *J. Bacteriol.* **183**, 4210-4216 (2001).
17. Chen X, *et al.* Structural identification of a bacterial quorum-sensing signal containing boron. *Nature* **415**, 545-549 (2002).
18. Colosimo DA, *et al.* Mapping interactions of microbial metabolites with human g-protein-coupled receptors. *Cell. Host. Microbe.* **26**, 273-282 e277 (2019).
19. Hawver LA, Jung SA, Ng WL. Specificity and complexity in bacterial quorum-sensing systems. *FEMS Microbiol. Rev.* **40**, 738-752 (2016).
20. Hense BA, Schuster M. Core principles of bacterial autoinducer systems. *Microbiol. Mol. Biol. Rev.* **79**, 153-169 (2015).
21. Wellington S, Greenberg EP. Quorum sensing signal selectivity and the potential for interspecies cross talk. *mBio* **10**, e00146-00119 (2019).
22. Pedregosa F, *et al.* Scikit-learn: Machine learning in python. *Journal of Machine Learning Research* **12**, 2825-2830 (2011).

Reviewers' Comments:

Reviewer #1:

Remarks to the Author:

The authors have addressed my remaining comments sufficiently.

Reviewer #2:

Remarks to the Author:

The authors have satisfactorily addressed my questions/comments. I now recommend publication.

Dear reviewers,

Machine Learning Aided Construction of the Quorum Sensing Communication Network for Human Gut Microbiota

Shengbo Wu, Jie Feng, Chunjiang Liu, Hao Wu, Zekai Qiu, Jianjun Ge, Shuyang Sun, Xia Hong, Yukun Li, Xiaona Wang, Aidong Yang, Fei Guo, and Jianjun Qiao.

We are very grateful for the positive comments from reviewers and editors, which have now all been carefully considered in the revision of our manuscript. In the revised manuscript, all the changes are marked in red.

The following is the point-to-point response to individual comment marked in blue.

REVIEWERS' COMMENTS

Response to Reviewer #1:

Reviewer #1 (Remarks to the Author):

The authors have addressed my remaining comments sufficiently.

Response:

Thanks for your interest and positive comment in this work.

Response to Reviewer #2:

Reviewer #2 (Remarks to the Author):

The authors have satisfactorily addressed my questions/comments. I now recommend publication.

Response:

Thanks for the good comment in our work.